# An indigenous *Saccharomyces uvarum* population with high genetic diversity dominates uninoculated Chardonnay fermentations at a Canadian winery

**Garrett C. McCarthy**[1], **Sydney C. Morgan**[1]\*, **Jonathan T. Martiniuk**[2], **Brianne L. Newman**[1], **Stephanie E. McCann**[1], **Vivien Measday**[2], **Daniel M. Durall**[1]

**1** Department of Biology, Irfigving K. Barber School of Arts and Sciences, The University of British Columbia, Kelowna, British Columbia, Canada, **2** Wine Research Centre, Faculty of Land and Food Systems, The University of British Columbia, Vancouver, British Columbia, Canada

☯ These authors contributed equally to this work.
¤ Current address: Sanford Consortium for Regenerative Medicine, University of California San Diego, La Jolla, California, United States of America
\* scmorgan@health.ucsd.edu

**Data Availability Statement:** All raw data, R scripts, software packages, versions, parameters, and primer sequences used in this study can be viewed at https://osf.io/j7rx8/. The QIIME2 artifact

## Abstract

*Saccharomyces cerevisiae* is the primary yeast species responsible for most fermentations in winemaking. However, other yeasts, including *Saccharomyces uvarum*, have occasionally been found conducting commercial fermentations around the world. *S. uvarum* is typically associated with white wine fermentations in cool-climate wine regions, and has been identified as the dominant yeast in fermentations from France, Hungary, northern Italy, and, recently, Canada. However, little is known about how the origin and genetic diversity of the Canadian *S. uvarum* population relates to strains from other parts of the world. In this study, a highly diverse *S. uvarum* population was found dominating uninoculated commercial fermentations of Chardonnay grapes sourced from two different vineyards. Most of the strains identified were found to be genetically distinct from *S. uvarum* strains isolated globally. Of the 106 strains of *S. uvarum* identified in this study, four played a dominant role in the fermentations, with some strains predominating in the fermentations from one vineyard over the other. Furthermore, two of these dominant strains were previously identified as dominant strains in uninoculated Chardonnay fermentations at the same winery two years earlier, suggesting the presence of a winery-resident population of indigenous *S. uvarum*. This research provides valuable insight into the diversity and persistence of non-commercial *S. uvarum* strains in North America, and a stepping stone for future work into the enological potential of an alternative *Saccharomyces* yeast species.

## Introduction

Modern winemaking often involves the inoculation of grape must with one or more commercial yeast strains, usually belonging to the dominant winemaking yeast species *Saccharomyces*

can be viewed at https://view.qiime2.org. For raw Illumina sequence data, please visit https://qiita.ucsd.edu and search for Study 12837.

**Funding:** This work was supported by a Natural Sciences and Engineerig Research Council of Canada (NSERC) Discovery Grant (RGPIN-2016-04261) to V.M., and an NSERC Collaborative Research and Development Grant (CRDPJ 514045-17) to D.M.D. and V.M.

**Competing interests:** The authors have declared that no competing interests exist.

*cerevisiae*. However, single-strain fermentations have been shown to produce less complex wines than fermentations conducted by multiple yeast species and strains [1–3]. Furthermore, the commercial *S. cerevisiae* strains used in these inoculated single-strain fermentations may be more aggressive than indigenous yeasts; indeed, commercial strains used previously at wineries have been identified entering and dominating uninoculated (spontaneous) fermentations in subsequent vintages [4–8]. The loss of indigenous yeast strains during the winemaking process can result in the reduction of regional character, because the local consortium of microorganisms can contribute to the expression of *terroir* in wine [3, 9]. In recent years, many winemakers have become interested in conducting uninoculated fermentations in an attempt to encourage a diversity of yeast species and strains to participate in alcoholic fermentation. Although vineyard-derived yeasts were originally thought to be the ones conducting uninoculated fermentations, a growing body of evidence has shown that uninoculated fermentations are actually conducted by winery-resident yeast strains [4, 7, 10, 11]. These yeasts may be of commercial or indigenous origin, but have established themselves as residents of the winery environment, and are capable of entering and conducting fermentations over multiple vintages.

Although most uninoculated fermentations at commercial wineries are conducted by strains of *S. cerevisiae* [4, 7, 10, 11], some wineries contain established populations of *Saccharomyces uvarum* that are able to conduct and complete alcoholic fermentation [12–15]. *S. uvarum* belongs to the *Saccharomyces sensu stricto* clade, and is the furthest relative from *S. cerevisiae* within this clade [16, 17]. The Holarctic *S. uvarum* population, which originated in the northern hemisphere [18–20], includes both natural *S. uvarum* strains isolated from the soil and bark of *Quercus* (oak) trees as well as industrial strains isolated from cider, beer, and wine fermentations. It is believed that the Holarctic population evolved alongside other *Saccharomyces* species, as 95% of Holarctic *S. uvarum* strains possess introgressed regions from *Saccharomyces eubayanus*, *Saccharomyces kudriavzevii*, and *S. cerevisiae* throughout their genome [18, 19]. However, the history of *S. uvarum* research is difficult to trace, because *S. uvarum* has had many names, and has even shared names with now distinct species. In the past, *S. uvarum* has been referred to as *Saccharomyces bayanus* var. *uvarum* [13, 14, 21, 22] or simply *Saccharomyces bayanus* [23, 24]. To complicate matters, many commercial *S. cerevisiae* strains have been marketed incorrectly as strains of *S. bayanus* [25]. However, *S. uvarum* is now known to be a pure species, distinct from *S. bayanus*, which itself is a hybrid of the pure species *S. uvarum* and *S. eubayanus* [16, 19, 26, 27].

*S. uvarum* is a cryotolerant yeast usually found in association with white wine fermentations in cool-climate wine regions [12, 13, 15, 28], but has also been associated with cider production [23, 24] and some traditional fermentations [29, 30]. During fermentation, *S. uvarum* produces lower levels of ethanol, acetic acid, and acetaldehyde, and higher levels of glycerol, succinic acid, malic acid, isoamyl alcohol, isobutanol, and ethyl acetate, as compared to *S. cerevisiae* [31–34]. Additionally, because of its ability to conduct fermentation at lower temperatures, *S. uvarum* may produce wines with more balanced aroma profiles [35]. However, few studies have been conducted to investigate the origins, genetic diversity, and enological potential of this yeast, and thus more research is needed on this topic.

The overall objective of this study was to identify the presence and genetic diversity of *S. uvarum* strains conducting uninoculated fermentations at a commercial winery in the Okanagan Valley wine region of British Columbia, Canada, and place this population within the context of global *S. uvarum* strains. Two years before this current study was conducted, a highly diverse, indigenous population of *S. uvarum* was identified at this same commercial winery [15]. We were interested in investigating the persistence of *S. uvarum* strains in the winery environment over multiple vintages. A secondary objective (addressed by sampling

fermentations with grapes sourced from two different vineyards) was to investigate whether the geographical origin or chemistry of the grapes played a role in determining the fungal communities and *S. uvarum* populations present in the fermentations.

## Materials and methods

### Experimental design and sampling

This study was conducted in association with Mission Hill Family Estate Winery during the 2017 vintage. The fungal diversity and community composition of grapes from two different vineyards were followed from the vineyard into the winery and throughout alcoholic fermentation. Samples for high-throughput amplicon sequencing (Illumina MiSeq) were taken from grapes in the vineyard just prior to harvest, as well as at four stages of fermentation in the winery. Samples for *Saccharomyces uvarum* population analysis were taken at three stages of alcoholic fermentation.

**Vineyard grape samples.** Two Chardonnay vineyards managed by Mission Hill Family Estate Winery in the Okanagan Valley of British Columbia, Canada, were selected for this study (Vineyard 2 and Vineyard 8). Vineyard 2 is located approximately 120 km south of the winery, and Vineyard 8 is located approximately 90 km south of the winery; the two vineyards are approximately 30 km apart. The exact locations of these vineyards, along with the dates of grape sample collection, can be viewed in Table 1. Both vineyards had been herbicide-free since 2016 (2017 was the second herbicide-free vintage), and were transitioning from conventional to organic viticulture practices.

Each vineyard was divided into six sampling sections (achieved by dividing the number of rows in the vineyard by six), and was further sub-divided into potential sampling sites within each section. Each post in the vineyard was planted approximately 15 feet (4.57 m) apart, with five vines planted in between each post (within each panel). A potential sampling site was defined as three consecutive panels (equalling approximately 15 vines), and one sampling site was randomly selected within each sampling section, for a total of six samples per vineyard (S1 Fig). The following restrictions were placed on randomization: sampling occurred at least three posts inward from the edge of each vineyard, and at least two rows away from neighbouring vineyard blocks, to minimize the potential for contamination from nearby roads or other grape varietals. Sampling was performed by aseptically collecting one cluster from each vine in the sampling site, on both sides of the row, for a total of 30 clusters per sample (approximately 2–3 kg). The 30 clusters from each sampling site were placed into a sterile bag (one bag per sampling site, six bags per vineyard) and transported on ice back to the laboratory at The University of British Columbia (Kelowna, BC, Canada) for same-day processing. Each bag containing grape clusters was then gently crushed and homogenized by hand for 10 min, after which time 2 mL samples of the juice were collected and frozen at -80˚C to await further processing for high-throughput amplicon sequencing.

**Winery samples.** The winery portion of this study was conducted at Mission Hill Family Estate Winery, a large commercial winery in British Columbia, Canada that conducts both inoculated and uninoculated (spontaneous) fermentations of many different grape varietals and makes wines in many different styles. The Chardonnay grapes for this study were sourced

**Table 1. Summary of vineyards used in this study.**

| Vineyard name | Closest municipality | Latitude | Longitude | Sample collection | Date of commercial harvest |
|---|---|---|---|---|---|
| Vineyard 2 | Osoyoos, BC | 49.000603 | -119.418712 | 2017-09-04 | 2017-09-05 |
| Vineyard 8 | Oliver, BC | 49.221125 | -119.559834 | 2017-09-18 | 2017-09-19 |

from two vineyards (Vineyard 2 and Vineyard 8), as described above. The must from each vineyard was crushed and pressed into large stainless steel tanks to undergo a settling period before being transferred into 285 L stainless steel barrels (La Garde, SML Stainless Steel, Québec, Canada). Must from Vineyard 2 spent two days in the settling tank, while must from Vineyard 8 spent eight days in the settling tank. Six stainless steel barrels were used in this experiment: three contained must exclusively from Vineyard 2, and three contained must exclusively from Vineyard 8. Because Vineyard 2 is located further south than Vineyard 8 (Table 1), the grapes from this vineyard ripened earlier, and were harvested before those from Vineyard 8, in order to achieve similar sugar concentrations at crush. Although the fermentations from each vineyard began at different times, they did overlap in the same cellar for 20 days during sampling. The cellar where the fermentations were conducted was maintained at 12˚C.

Samples were taken from the stainless steel barrels at four stages of uninoculated (spontaneous) alcoholic fermentation as defined by their sugar concentration (approximated by Brix level): cold settling (22˚Brix, prior to the start of AF), early (14–18˚Brix), mid (6–10˚Brix), and late (2 ± 0.1˚Brix). To each barrel, 20 mg/L total sulfur dioxide was added in the form of potassium metabisulfite ($K_2S_2O_5$) during the cold settling stage, 250 ppm Lallemand® Fermaid K complex yeast nutrient was added between the cold settling and early stages, and 100 ppm Laffort® THIAZOTE mineral nutrient was added between the early and mid stages. Samples were collected in sterile 50 mL centrifuge tubes and were transported on ice to the laboratory at The University of British Columbia (Kelowna, BC, Canada) for processing. Subsamples (2 mL) were washed and frozen at -80˚C for high-throughput amplicon sequencing, while other subsamples were processed immediately for culture-dependent *S. uvarum* strain typing.

## Chemical analyses

Chemical parameters for the grape must and wine were taken from the stainless steel barrels at the cold settling and late stages, respectively. Yeast assimilable nitrogen (YAN), titratable acidity, volatile acidity, malic acid, pH, residual sugar, ethanol content, fructose, and glucose were measured using a WineScan™ wine analyzer (Foss, Hillerød, Denmark).

## *Saccharomyces uvarum* strain typing

**Colony isolation and DNA extraction.** Samples from the early, mid, and late stages of fermentation were serially-diluted and plated onto YEPD media (1% (w/w) yeast extract (BD Bacto™, Sparks, MD, USA), 1% (w/w) bacterial peptone (HiMedia Labs, Mumbai, India), 2% (w/w) dextrose (Fisher Chemical, Fair Lawn, NJ, USA), 2% (w/w) agar (VWR, Solon, OH, USA)), with the addition of 0.01% (v/v) chloramphenicol (Sigma-Aldrich, St. Louis, MO, USA) to prevent bacterial growth, and 0.015% (v/v) biphenyl (Sigma-Aldrich, St. Louis, MO, USA) to prevent filamentous fungal growth [12, 36–38]. Plates were incubated at 28˚C for 48 h and stored at 4˚C. Plates containing 30–300 colonies were selected for colony isolation and DNA extraction. Grape samples from the vineyard and the cold settling stage samples were not analyzed for *S. uvarum* strains, because *Saccharomyces* yeasts are rarely found in must before the onset of alcoholic fermentation [39], and are present on healthy grapes in very low abundance compared to other microbes [40].

From the early, mid, and late stage plates, 47 single yeast colonies were isolated onto Wallerstein Nutrient media (WLN) (Sigma-Aldrich, St. Louis, MO, USA), a differential medium used to identify non-*S. cerevisiae* yeasts. Two controls were used to distinguish between *S. cerevisiae* and *S. uvarum* colonies: Lalvin® BA11 (Lallemand, Montreal, QC, Canada) and CBS 7001 (Westerdijk Fungal Biodiversity Institute, Utrecht, Netherlands), respectively. The WLN

plates were incubated at 28˚C for 48 h, and then stored at 4˚C. Presumed *S. cerevisiae* isolates appeared cream-coloured, while presumed *S. uvarum* isolates appeared green (S2A Fig). In order to determine the colour of potential *S. cerevisiae* x *S. uvarum* hybrids on WLN plates, the following isolates were streaked onto a WLN plate and incubated at 28˚C for 48 h: a pure *S. cerevisiae* strain (Lalvin® BA11), a pure *S. uvarum* strain (CBS 7001), and hybrid of *S. cerevisiae* x *S. uvarum* (Lalvin® S6U, formerly referred to as a *S. cerevisiae* x *S. bayanus* hybrid) (S2B Fig). DNA was extracted from each yeast isolate by performing a water DNA extraction, as described previously [7].

**Multiplex PCR and fragment analysis.** *S. uvarum* strain typing was performed as previously described [15] using a multiplex PCR reaction targeting 11 microsatellite loci that had been selected from two previous studies: L1, L2, L3, L4, L7, L8, L9 [41], NB1, NB4, NB8, and NB9 [28]. Briefly, multiplex PCR was performed on the extracted DNA from each presumed *S. uvarum* isolate and submitted to Fragment Analysis and DNA Sequencing Services at the University of British Columbia (Kelowna, BC, Canada) for fragment analysis on a 3130xl DNA sequencer (Applied Biosystems, Foster City, CA, USA). GeneMapper 4.0 software (Applied Biosystems, Foster City, CA, USA) was used to determine the fragment size at each locus, and the resulting multilocus genotype (MLG) of each isolate was compared to that of the others using Bruvo's genetic distance [42]. Yeast isolates obtained in this study were compared to a database containing 150 *S. uvarum* strains identified during the 2015 vintage at the same winery [15], as well as 12 international *S. uvarum* strains (Table 2).

Bruvo distance was calculated using an algorithm that takes into account stepwise mutations, making it appropriate for use with microsatellite data. Bruvo distance is calculated on experimental data and allows the user to collapse MLGs with slight differences in allele size into a single strain category, based on similarity at a threshold value from 0 to 1. An applied threshold of 0 results in every unique MLG being classified as a different strain, and an applied threshold of 1 results in all the MLGs in a dataset being collapsed into a single strain. Bruvo distance was calculated in R (version 3.5.1) using the 'poppr' package (version 2.8.1) [43], and applying a genetic distance threshold of 0.3 for strain classification. A histogram of pairwise genetic distances was created in R (version 3.5.1) using the 'poppr' package (S3 Fig). Because the histogram was not bimodal, the largest gap between putative thresholds was determined visually as 0.3 [44]. Additionally, 0.3 was chosen as the cut-off threshold in order to remain consistent with the threshold used in a previous study of this same population [15]. This was particularly important because of the novelty of the high genetic diversity observed in this

**Table 2. Names, geographical origins, and sources of international *Saccharomyces uvarum* strains used for comparison in this study.**

| Name | Geographical Origin | Source |
|---|---|---|
| CBS 395 | The Netherlands | Centraalbureau voor Schimmelcultures (CBS) |
| CBS 7001 | Spain | |
| CBS 8690 | Moldova | |
| CBS 8696 | California (USA) | Westerdijk Fungal Biodiversity Institute (Utrecht, The Netherlands) |
| CBS 8711 | France | |
| PYCC 6860 | Hornby Island (Canada) | Portuguese Yeast Culture Collection (PYCC) |
| PYCC 6861 | Hornby Island (Canada) | |
| PYCC 6862 | Japan | |
| PYCC 6871 | Portugal | Universidade Nova de Lisboa (Caparica, Portugal) |
| PYCC 6901 | Oregon (USA) | |
| PYCC 6902 | Missouri (USA) | |
| Velluto BMV58® | Spain | Commercial strain from Lallemand® |

yeast population. Using a low cut-off threshold for distinguishing strains could overestimate this diversity, and artificially inflate the significance of our results; by using a conservative threshold, we are able to make more confident statements regarding the diversity of this population.

Isolates that only partially amplified were re-run and were subsequently excluded from analysis upon a second failure. Isolates that did not amplify were considered to be non-*S. uvarum* yeasts and were also excluded from analysis. Although 47 yeast isolates were originally selected from each sample, not all belonged to *S. uvarum* and some were presumed to be *S. cerevisiae* based on WLN. After strain identification, each sample was rarefied to 32 *S. uvarum* isolates.

Presumed *S. cerevisiae* isolates were also strain-typed as described previously [15] in order to confirm their identities, but were not included in the analysis of this study.

**RFLP analysis for hybrid investigation.** Restriction fragment length polymorphism (RFLP) analysis was performed on 50 yeast isolates from this study, representing the most abundant 50 strains identified. A pure *S. cerevisiae* strain (Fermol® Mediterranée), a pure *S. uvarum* strain (CBS 7001), and a known *S. cerevisiae* x *S. uvarum* hybrid strain (Lalvin® S6U) were included in this analysis as reference strains. PCR-RFLP was performed via amplification of the ITS1 region of the rRNA gene using primers ITS1 and ITS4, followed by digestion using the restriction enzyme *Hae*III (Product no. ER0151, Thermo Fisher, Waltham, MA, USA), as described previously [45]. The products were run on a 1.8% agarose gel and the resulting patterns of each of the 50 strains were compared to those of the reference strains (S4 Fig).

## High-throughput amplicon sequencing

**Sample treatment and DNA extraction.** Samples for high-throughput amplicon sequencing (Illumina MiSeq) were taken from grapes in the vineyard (grapes), as well as from four stages of fermentation in the winery (cold settling, early, mid, and late).

Samples (previously frozen at -80˚C) were thawed on ice, and then washed before total DNA was extracted following a modified protocol from a previously-published study [46]. Samples were pelleted by centrifugation at 13,200 rpm for 5 min. The supernatant was discarded and the pellet was re-suspended in 1 mL chilled 1x phosphate-buffered saline (Sigma-Aldrich, St. Louis, MO, USA), then centrifuged again at 13,200 rpm for 5 min. The supernatant was discarded and the pellet was re-suspended in 500 μL of 50 mM ethylenediaminetetraacetic acid (EDTA, pH 8.0) (Invitrogen, Grand Island, NY, USA). Samples were mixed by pipetting up and down five times, and the entire sample was transferred to a FastPrep tube (MP Biomedicals, Santa Ana, CA, USA) containing 200 mg of 0.5 mm glass disruptor beads (Scientific Industries, Bohemia, NY, USA). The FastPrep tubes were placed into a Vortex-Genie 2 Digital bead beater (Scientific Industries, Bohemia, NY, USA) for two rounds of 2.5 min (30/s), separated by 1 min on ice. Aliquots of 500 μL Nuclei Lysis solution (Fisher, Hampton, VA, USA) were added to the FastPrep tubes and lysed in the bead beater for 1 min (30/s). Samples were then incubated for 10 min at 95˚C and then centrifuged at 13,200 rpm for 5 min. To an autoclaved 2 mL microcentrifuge tube (VWR, Radnor, PA, USA), 500 μL of the supernatant was added, followed by 250 μL Protein Precipitation solution (Fisher, Hampton, VA, USA). Samples were then vortexed lightly and kept at room temperature (22˚C) for 15 min. Samples were then centrifuged at 13,200 rpm for 5 min, and 500 μL of the supernatant was transferred to new 2 mL microcentrifuge tubes containing 75 μL 20% (v/v) polyvinylpyrrolidone (PVP) solution (Sigma-Aldrich, St. Louis, MO, USA). Samples were pulse-vortexed for 10–20 sec and centrifuged at 13,200 rpm for 10 min. Supernatant (500 μL) was transferred to a new 2 mL microcentrifuge tube containing 300 μL chilled 2-propanol (Sigma-Aldrich, St. Louis, MO,

USA). Tubes were inverted to mix several times and left to sit at room temperature for 15 min. Samples were then centrifuged at 13,200 rpm for 2 min and the supernatant was discarded. The pellet was re-suspended in 1 mL chilled 100% ethanol (Commercial Alcohols, Brampton, ON, Canada), centrifuged at 13,200 for 2 min, and the entire supernatant was carefully discarded. Samples were left open in a biosafety cabinet for a maximum of 30 minutes to ensure complete evaporation of the ethanol. The samples were re-suspended in 50 μL of 10 mM TE buffer (Invitrogen, Grand Island, NY, USA), and frozen at -80˚C. A positive control, containing pure *S. cerevisiae* cells, and a negative control, containing only molecular-grade water, were also subjected to the same DNA extraction protocol, and were included during library preparation and Illumina sequencing.

**Illumina MiSeq library preparation.** Sample library preparation used a two-step PCR procedure consisting of 'amplicon' and 'index' PCR reactions, as described previously [47]. Amplicon PCR was performed by amplifying the ITS1 region of the rRNA gene using BITS and B58S3 primers [48] with CS1 and CS2 linker sequences, respectively. Index PCR primers contained Illumina MiSeq adapter sequences, unique eight nucleotide barcodes, 9–12 bp heterogeneity spacers, and CS1/CS2 linker sequences. After both PCR reactions, samples were submitted to the IBEST Genomics Resources Core at the University of Idaho (Moscow, ID, USA) for quantification, normalization, pooling, and sequencing. Paired-end sequencing (300 bp) was performed on an Illumina MiSeq Desktop Sequencer (Illumina Inc., San Diego, CA, USA).

**Illumina MiSeq data processing.** Illumina MiSeq data processing was performed using both R (version 3.5.1) and the open-source bioinformatics pipeline Quantitative Insights Into Microbial Ecology (QIIME2 version 2018.11) [49]. In R, sequences were denoised using the 'dada2' package (version 1.8) [50], as well as the 'ShortRead' (version 1.36.1) [51] and 'Biostrings' (version 2.46.0) packages. All forward and reverse primer sequences had been removed from the 5' end of the sequences by IBEST at the University of Idaho before being returned, but because some ITS sequences are likely to be shorter than 300 bp, it is possible that these sequences contain nucleotides from the opposite primer, which required removal before further processing using Cutadapt [52]. After all primer sequences had been removed, sequences were filtered and trimmed using the "filterandTrim" function in the 'dada2' package. Any sequence containing an N was removed, as well as any sequence shorter than 50 bp. The maximum number of expected errors allowed in any sequence was set to 2. Because the ITS1 region in fungi is highly variable [53, 54], trimming all sequences to the same length can reduce the diversity of the identified community and can even remove sequences with true lengths shorter than the specified truncation length. For this reason, sequences were not trimmed to a consistent length. Forward and reverse reads were merged using the "mergePairs" function, a sequence table was constructed with the "makeSequenceTable," and chimeras were removed with the "removeBimeraDenovo" function. The representative sequence table was converted to a Fasta file before being transferred from R to the QIIME2 pipeline to complete analysis.

In QIIME2, sequences underwent paired-end alignment using MAFFT [55], and a phylogenetic tree with a mid-point root was created using FastTree 2 [56]. Sequence variants were classified to the species level (if possible) using a dynamic (97–99%) threshold classifier made with the UNITE (version 8.0) database [57]. Sequence variants that could not be classified to the order level or lower and those that appeared with a total frequency of < 100 sequences were excluded from analysis. Samples were rarefied to 20,000 sequences before being exported from QIIME2 for statistical analysis and visualization. Two samples did not meet the applied threshold of 20,000 sequences, and were therefore removed from analysis: one sample was a cold settling sample from the Vineyard 8 fermentations, and the other was a grape sample from Vineyard 2. In the UNITE (version 8.0) database, *S. uvarum* is incorrectly classified as *S. bayanus*, because in the past both species were considered synonymous. Based on our culture-

dependent data, we are confident that sequences classified in the UNITE database as *S. bayanus* belong to *S. uvarum*, and we have accordingly re-named all our sequences identified as *S. bayanus* to *S. uvarum*.

## Statistical analysis

All statistical analyses in this study, unless otherwise specified, were performed in R (version 3.5.1) using RStudio software, and all statistical tests assume a significance level of $\alpha = 0.05$. The chemical parameters of the grape must and wine from each vineyard were statistically compared by performing a one-way analysis of variance (ANOVA), using the "aov" function. Each parameter was evaluated separately. Normality was assessed visually. Not all of the chemical parameters met the assumption of normal distribution, but as ANOVA are robust to departures from normality [58], no data transformations were performed. The assumption of homogeneity of variance was tested using the "leveneTest" function in the 'Rcmdr' package (version 2.5–1). Levene's test indicated no violation in the assumption of homogeneity of variance in any chemical parameter (S1 Table).

Rarefaction curves of both species richness (for the fungal community) and strain richness (for the *S. uvarum* population) were created using the "rarecurve" function in the 'vegan' package (version 2.5–1) (S5 Fig) [59]. The fungal community was rarefied to 20,000 sequences per sample; species richness reached a plateau at this sequencing depth for all the samples that were retained after rarefying (S5A Fig). The *S. uvarum* population did not reach terminal sampling depth before the rarefaction point of 32 isolates per sample (S5B Fig); however, this number was chosen because it was the highest number of *S. uvarum* isolates that allowed every sample to be retained for analysis.

Simpson's Index of Diversity $(1 − D)$ and Shannon's Diversity Index $(H)$ were calculated using the "diversity" function in the 'vegan' package (version 2.5–1) and reported ± the standard error of the mean (SEM). Differences in diversity between the two vineyard treatments were assessed by performing a one-way ANOVA on grape samples using the "aov" function, and by performing one-way repeated-measures ANOVA on both the fungal communities and the *S. uvarum* populations throughout alcoholic fermentation, using the "Anova" function in the 'car' package (version 3.0–2). Grape samples were analyzed separately from the fermentation samples because they represent a different sample type, and cold settling samples were excluded from analysis because Vineyard 8 contained only two replicates at that stage. Residuals and histograms were plotted to test the assumptions of the model.

The relative abundance of fungal species and *S. uvarum* strains was visualized by creating stacked bar charts using GraphPad Prism (version 8.2.1) software (La Jolla, CA, USA). Differences in composition between vineyard treatments were assessed by performing permutational analysis of variance (PERMANOVA) tests using the "adonis" function (vegan package), and using Bray-Curtis dissimilarity matrices, which were calculated on untransformed abundance data using the "vegdist" function (vegan package). The assumption of homogeneity of multivariate group dispersions (PERMDISP) was analyzed using the "betadisper" and "permutest" functions (vegan package), using Bray-Curtis dissimilarity and calculating deviation from centroid. Test statistics (*F* values for PERMDISP and Pseudo-*F* values for PERMANOVA) were calculated based on 999 permutations of raw data. No violation of the assumption of homogeneity of multivariate group dispersions was observed for the grape samples ($F(1,9) = 0.65$, $p = 0.51$) or the *S. uvarum* population during alcoholic fermentation ($F(1,16) = 1.9$, $p = 0.19$). The fungal community during alcoholic fermentation did violate this assumption ($F(1,16) = 10.4$, $p = 0.001$), but PERMANOVA tests are considered robust to unequal variances among treatments [60], so no data transformations were made. With regards to the fungal

community, the grape samples were analyzed separately from the fermentation samples because they represent a different sample type, and the cold settling samples were excluded from analysis because Vineyard 8 contained only two biological replicates at that stage.

A principal coordinates analysis (PCoA) was generated using the "wcmdscale" and "ordi-hull" functions (vegan package), using Bray-Curtis dissimilarity, in order to visualise the spatial distribution of the *S. uvarum* population among samples and treatments. The design of this study involved repeated measures, meaning some data were not independent, which could potentially lead to an overestimation of treatment differences as a result of the PERMANOVA tests, which cannot account for repeated measures. Therefore, the PCoA ordination was used to visualize distances between samples of different treatments [9, 61]. Finally, we note that because the fermentations from both vineyards did not occur simultaneously (Vineyard 8 was harvested 14 days after Vineyard 2), and because the Vineyard 8 must spent longer in the large settling tank, it is possible that the differences in fungal communities and *S. uvarum* populations observed in this study are not solely a result of differences in vineyard geography.

*S. uvarum* population structure was assessed on the 102 indigenous strains identified in this study (36 unique to 2017, 66 found in both 2015 and 2017) [15], as well as 12 international *S. uvarum* strains from ten geographic locations, isolated from winery and natural environments around the world (Table 2), for a total of 114 strains. Population structure was assessed by performing Bayesian clustering in InStruct [62], using the admixture model with a burn-in of 500,000 and a total run of 100,000 iterations with 5 chains per cluster (K), from K = 1 to K = 12. The 12 clusters were made up of the two vintages analyzed at this winery as well as the 10 geographic origins of the 12 international strains. The Deviance Information Criterion (DIC) method, outlined previously [63], was used to determine the optimal number of sub-populations, which was found to be K = 11. A plot of DIC at the last chain of 5, according to K revealed an additional, minor minimum plateau at K = 5, thus K = 5 was also included and passed onto CLUMPP for alignment of the chains. Five InStruct chains at K = 11 and K = 5 were aligned using the *LargeKGreedy* algorithm in CLUMPP (version 1.1.2) [64], with 10,000 random input orders. The highest H value derived from the CLUMPP population alignment was 0.51 for K = 11 and 0.99 for K = 5. The higher H value for the K = 5 clusters suggested better CLUMPP-alignment of the InStruct output than for K = 11, although the DIC value recommended the optimal number of clusters was 11. Therefore, inferred ancestry profiles were visualized for both K = 11 and K = 5 subpopulations. The CLUMPP-aligned ancestry profiles were visualized using DISTRUCT (version 1.1) [65], which provides a stacked bar plot for each strain, with strains (bars) partitioned into coloured segments that correspond to membership coefficients of inferred subpopulations. ObStruct [66] was used to determine significance of the InStruct-inferred population structure. GenAlEx (version 6.5) [67, 68] was used to calculate the Probability of Identity ($P_I$) and fixation indices ($F_{ST}$ and $F_{IS}$) for the *S. uvarum* population observed in this study, and to estimate heterozygosity.

An unrooted, neighbour-joining phylogenetic tree was generated to compare the genetic relatedness of the indigenous *S. uvarum* strains and the 12 international strains. The phylogenetic tree was generated using the 'ape' package (version 5.2) [69], while clustering of strains was accomplished by using the Bayesian clustering output of InStruct (K = 5 clusters) for statistical grouping of subpopulations in the tree. A dominant ancestor was identified if the inferred coefficient was equal to or higher than 0.75 (75%), based on the metric used previously for a similar population analysis [41]. Strains were coloured according to their dominant inferred ancestor. Bootstrap values were obtained using the "bruvo.boot" function in the 'poppr' package (version 2.8.1) [43], which randomly sampled loci 1000 times to recalculate the percent support of the tree success. Only branches with 50% support or higher were indicated on the tree.

**Table 3. Chemical parameters of stainless steel barrel-fermented Chardonnay wines sourced from two vineyards.** Samples were taken at the cold settling stage (prior to the start of alcoholic fermentation) and at the late stage (towards the end of alcoholic fermentation). Values are the mean ± SEM ($n$ = 3 barrels per vineyard treatment). An asterisk next to the chemical parameter indicates a significant difference ($p \leq 0.05$) between the two vineyards. Each chemical parameter was evaluated separately within each stage.

| Stage | Chemical Parameter | Vineyard 2 | Vineyard 8 |
|---|---|---|---|
| Cold settling | pH* | 3.37 ± 0.006 | 3.31 ± 0.003 |
| | Residual sugar (˚Brix)* | 22.3 ± 0.0 | 22.0 ± 0.03 |
| | Yeast assimilable nitrogen (mg/L)* | 77.7 ± 5.6 | 110.7 ± 3.9 |
| | Titratable acidity (g/L)* | 5.07 ± 0.03 | 7.20 ± 0.0 |
| | Malic acid (g/L)* | 2.20 ± 0.0 | 3.47 ± 0.03 |
| Late | pH | 3.37 ± 0.01 | 3.41 ± 0.01 |
| | Titratable acidity (g/L)* | 7.20 ± 0.2 | 7.83 ± 0.03 |
| | Malic acid (g/L)* | 1.97 ± 0.03 | 2.40 ± 0.06 |
| | Volatile acidity (g/L)* | 0.47 ± 0.003 | 0.38 ± 0.001 |
| | Ethanol (% v/v) | 11.7 ± 0.1 | 11.3 ± 0.1 |
| | Glucose (g/L) | 1.07 ± 0.1 | 1.37 ± 0.3 |
| | Fructose(g/L) | 28.9 ± 1 | 27.1 ± 2 |

All raw data and R scripts used in the preparation of this manuscript can be viewed at https://osf.io/j7rx8/.

## Results and discussion

### Fermentation progression and wine chemistry

The total time from harvest of the Chardonnay grapes to the end of alcoholic fermentation was similar for the must from both vineyards: the must from Vineyard 2 completed fermentation 40 days after harvest, and the must from Vineyard 8 completed fermentation 37 days after harvest. However, the time the must spent in the settling tank before being transferred to barrels for fermentation differed between the two vineyard treatments. Must from Vineyard 2 spent only two days in the large stainless steel settling tank, while must from Vineyard 8 spent eight days, due to a higher proportion of solid material that required a longer settling time.

Sugar concentration at crush was similar between the two vineyards (Table 3). Although a significant difference was found between the vineyards for sugar concentration (˚Brix), this is likely because of the low variation observed within each treatment. It is unlikely that a difference of 0.3˚Brix would have any biologically relevant effects on the microbial composition of the wine. The pH was higher in the must from Vineyard 2, while yeast assimilable nitrogen, titratable acidity, and malic acid were all significantly higher in the must from Vineyard 8.

By the late stage of alcoholic fermentation, there was no significant difference between the wines from the two vineyards in terms of pH, ethanol content, or glucose and fructose concentration (Table 3). Very little glucose remained in the wine by the end of alcoholic fermentation (< 2 g/L), while close to 30 g/L fructose remained unfermented.

By the late stage, titratable acidity and malic acid were still significantly higher in the wines from Vineyard 8 than the wines from Vineyard 2 (Table 3). However, the change in malic acid from the cold settling to the late stage was also much greater in the wines from Vineyard 8. In the wines from Vineyard 2, the malic acid concentration decreased from 2.20 g/L to 1.97 g/L, a decrease of approximately 10%. Meanwhile, in the wines from Vineyard 8, the malic acid concentration changed from 3.47 g/L to 2.40 g/L, a decrease of approximately 30%. This could suggest a more significant presence of malic acid-degrading bacteria in the must from Vineyard 8. Indeed, a previous study conducted at this same winery with grapes from Vineyard 8

identified *Tatumella* spp. in barrels that did not receive $SO_2$ [15]. These bacteria were thought to originate in the vineyard, as they were present in vineyard samples, but seemed to be sensitive to $SO_2$; the bacteria were not able to persist in treatments that received 40 mg/L $SO_2$ at crush. When they did persist, however, the malic acid was almost completely degraded by the end of alcoholic fermentation. It is possible that the addition in this current study of only 20 mg/L $SO_2$ allowed a portion of these bacteria to survive in the must and perform a partial degradation of malic acid. Unfortunately, we did not identify the bacterial community in this current study, so more research is needed to confirm this hypothesis.

Volatile acidity (estimated as acetic acid) was found to be significantly higher in the wines from Vineyard 2 (Table 3). However, neither vineyard produced wines with unacceptable levels of volatile acidity, and all the wines in this study contained volatile acidity levels that were below its sensory detection threshold of 0.7 g/L [70].

## Fungal communities

**Fungal community diversity.** Fungal species diversity was highest in the grape and cold settling samples for both vineyards, and decreased for the early, mid, and late stages of fermentation (Table 4). A decrease in the diversity of fungal taxa is expected at the onset of alcoholic fermentation, because most yeasts and fungi present on grapes are non-fermentative, and are either killed by the presence of ethanol or absence of oxygen, or out-competed for space and nutrients by the more dominant yeasts [39, 71]. Diversity did not change throughout the three alcoholic fermentation stages (early, mid, and late) within each vineyard treatment, but the wines from Vineyard 8 had a consistently higher diversity than the wines from Vineyard 2, regardless of the diversity index used.

**Fungal community composition.** In total, 194 fungal species were identified in this study, of which 11 achieved $\geq$ 10% relative abundance in at least two samples (S2 Table). These 194 species belonged to 19 different classes and 37 different orders; however, only four classes (Dothideomycetes, Eurotiomycetes, Leotiomycetes, Saccharomycetes), and six orders (Capnodiales, Dothideales, Pleosporales, Eurotiales, Erysiphales, Saccharomycetales), were represented in the top 90% of all identified sequences. For a list of all species identified in this study along with their taxonomic classifications, please visit https://osf.io/j7rx8/.

A PERMANOVA was performed to test the differences in community composition between grape (vineyard) samples, which were found to be significantly different ($F(1,9) = 8.3$, $R^2 = 0.48$, $p = 0.002$). Interestingly, the fungal communities of the grape and cold settling samples within each vineyard treatment were also different from each other (Fig 1). The cold settling samples were taken after the grapes had been harvested, crushed, and processed, so at

**Table 4. Fungal species diversity, measured as Simpson's Index of Diversity ($1 - D$) and Shannon's Diversity Index ($H$), of stainless steel barrel-fermented Chardonnay sourced from two different vineyards.** Diversity ± SEM was measured from grape samples taken in the vineyard ($n = 5$ for Vineyard 2, $n = 6$ for Vineyard 8), as well as at four stages in the winery: cold settling ($n = 3$ for Vineyard 2, $n = 2$ for Vineyard 8), early, mid, and late ($n = 3$ for both vineyards at all three fermentation stages). For each diversity index, a one-way repeated-measures ANOVA was performed to compare the differences between vineyards across the three fermentation stages (cold settling samples were not included because Vineyard 8 contained only two replicates). Grape samples were analyzed separately by performing one-way ANOVA, because they constituted a different sample type. The *p*-values for each index are indicated in the appropriate columns, and any significant differences ($p \leq 0.05$) are in bold.

| | Simpson's diversity ($1-D$) | | | Shannon's diversity ($H$) | | |
|---|---|---|---|---|---|---|
| **Sample** | **Vineyard 2** | **Vineyard 8** | ***p* =** | **Vineyard 2** | **Vineyard 8** | ***p* =** |
| Grapes | 0.85 ± 0.03 | 0.69 ± 0.05 | **0.02** | 2.51 ± 0.20 | 0.69 ± 0.13 | **0.005** |
| Cold settling | 0.64 ± 0.03 | 0.87 ± 0.02 | | 1.65 ± 0.14 | 2.29 ± 0.15 | |
| Early | 0.06 ± 0.01 | 0.21 ± 0.09 | **0.01** | 0.17 ± 0.01 | 0.38 ± 0.14 | **0.006** |
| Mid | 0.06 ± 0.01 | 0.31 ± 0.04 | | 0.14 ± 0.02 | 0.61 ± 0.08 | |
| Late | 0.06 ± 0.01 | 0.30 ± 0.10 | | 0.15 ± 0.02 | 0.57 ± 0.20 | |

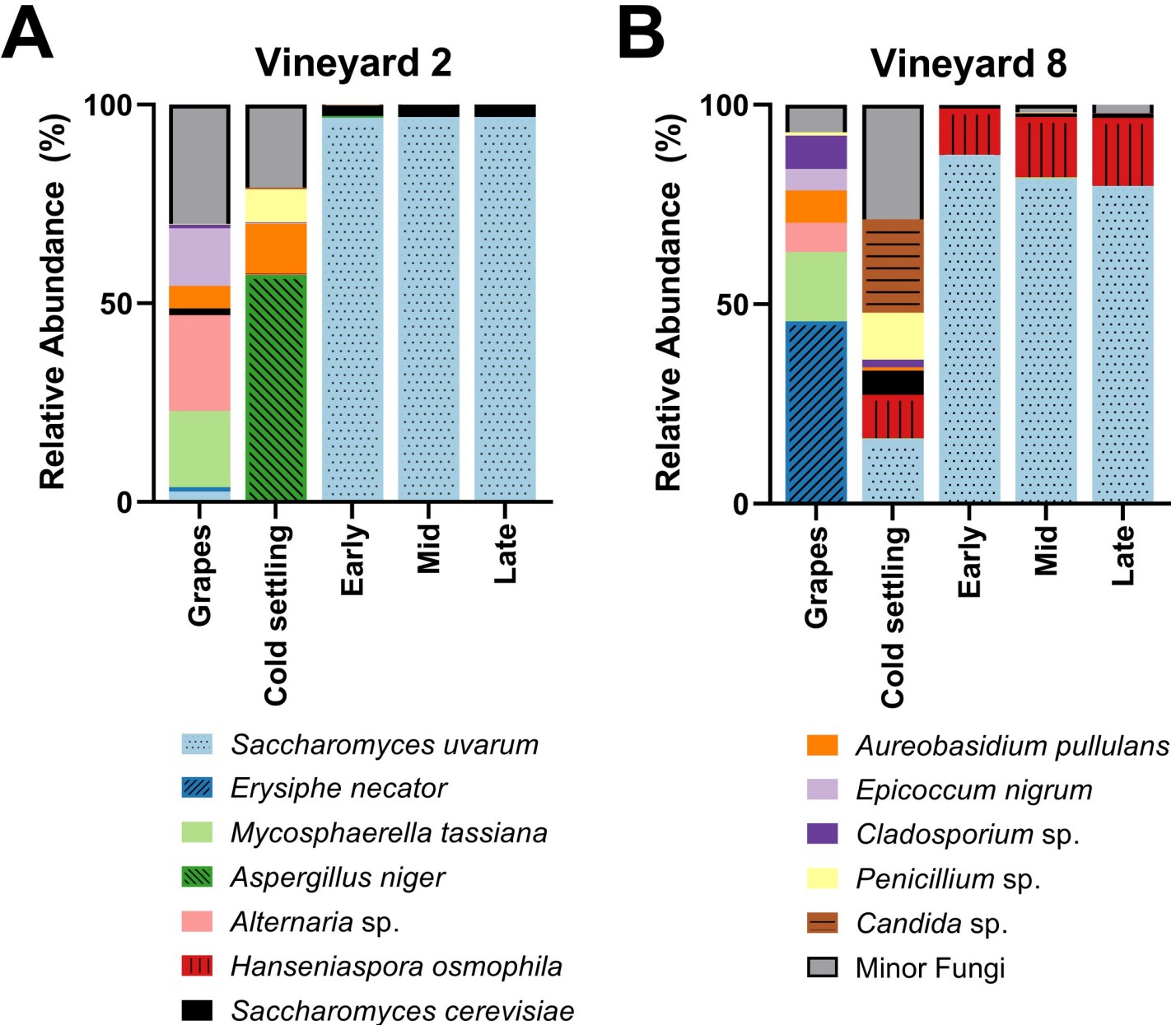

**Fig 1. Fungal species abundance.** Relative abundance of the dominant fungi present in grape samples taken in the vineyard (grapes), as well as at four stages of fermentation in the winery (cold settling, early, mid, and late), of Chardonnay sourced from (A) Vineyard 2 or (B) Vineyard 8. Vineyard 2 grape sample values are the means of five replicates, and Vineyard 8 grape sample values are the means of 6 replicates. All winery fermentation stages have three reported replicates, with the exception of the cold settling stage from the Vineyard 8 fermentations, which contained two. Relative abundance was calculated from 20,000 sequences per sample. Any fungal taxa that did not achieve at least 10% relative abundance in at least two samples were grouped into the Minor Fungi category, with one exception: *S. cerevisiae* did not achieve 10% in any one sample, but because of its importance during alcoholic fermentation it has been included here. For variation among samples please see S2 Table.

least some of the differences observed between these stages can be attributed to contact with winery equipment. In the Vineyard 2 treatment, the most abundant fungi in the grape samples were *Alternaria* sp., *Mycosphaerella tassiana*, and *Epicoccum nigrum*, while the most abundant fungi in the cold settling samples were *Aspergillus niger*, *Aureobasidium pullulans*, and *Penicillium* sp. In the Vineyard 8 treatment, the most abundant fungi in the grape samples were

*Erysiphe necator* and *Mycosphaerella tassiana*, while the most abundant fungi in the cold settling samples were *Candida* sp., *Penicillium* sp., *S. uvarum*, and *Hanseniaspora osmophila*.

*A. pullulans* is a ubiquitous environmental yeast-like fungus that is commonly associated with grapes and vineyards [47, 72, 73], and was also the most common fungal species identified at the cold settling stage in must from Vineyard 8 two years prior to this current study [15]. *E. nigrum* is an endophytic fungus that has been previously identified in wine regions such as Italy, Portugal, and Spain [74–76]. Because both *A. pullulans* and *E. nigrum* are typically associated with vineyard environments, they have been proposed as potential biological control agents against grapevine pathogens [77]. *Alternaria* sp. are plant pathogens that are also commonly identified on grapes and in grape must [47, 78, 79]. *M. tassiana*, also known as *Davidiella tassiana*, is a common grape symbiont that has been previously isolated from vineyards [15, 47, 80, 81], and has been identified as the most abundant fungal species isolated on grapes in one study [79]. The presence of *Aspergillus* and *Penicillium* spp. in food, including grape must, has been observed previously [15, 47, 82]. These fungi have the potential to produce mycotoxins, which can be dangerous if consumed in large quantities [83, 84]. However, low levels of mycotoxins are common in commercial wines [84], and Canadian wines typically have a lower concentration of mycotoxins than wines produced elsewhere in the world [85]. Furthermore, the process of alcoholic fermentation has been shown to reduce the presence of mycotoxins, through enzymatic conversion to a less toxic form and/or the adsorption to lees and subsequent removal from the wine [83, 86]. *E. necator* (also called *Uncinula necator*) is a grapevine pathogen responsible for powdery mildew, and is found in all regions of the world where grapes are grown [87]. The dominant presence of this fungus in the grape samples of Vineyard 8 is of potential concern, because the presence of even small amounts can have a negative effect on the overall sensory profile of the wine [88]. However, the grapes sampled in this study were not visibly infected with powdery mildew, and the presence of *E. necator* seems to have been eliminated by the time the cold settling sample was taken (Fig 1B). In the Vineyard 8 cold settling sample we observed the presence of yeasts with fermentative potential such as *Candida* sp., *H. osmophila*, and *S. uvarum*. These fermentative yeasts were not identified in the Vineyard 2 cold settling sample, likely because the must from Vineyard 8 spent an extended amount of time in the settling tank before being transferred to barrels (where the cold settling samples were taken), and therefore was more susceptible to early winery-resident yeast exposure. Although *Candida* and *Hanseniaspora* spp. have been found in association with grapes in the vineyard [89], in this current study it seems more likely that the origin of these yeasts is the winery equipment that the grapes came into contact with, because neither *Candida* nor *Hanseniaspora* spp. were identified in the grape samples from Vineyard 8 (S2 Table). The grape samples from Vineyard 2 had a greater presence of *Saccharomyces* spp. than the grape samples from Vineyard 8: *S. cerevisiae* was present at 1.61 ± 1.3% and *S. uvarum* at 2.61 ± 2.6% in Vineyard 2, and at 0.053 ± 0.03% and 0.027 ± 0.02%, respectively, in Vineyard 8 (S2 Table). The abundance of *Saccharomyces* yeasts on healthy grapes in vineyards had been estimated at approximately 1/1000 yeast isolates [40], which is more in line with the results observed in Vineyard 8. The increased abundance of *Saccharomyces* yeasts in Vineyard 2 should be further investigated.

A PERMANOVA was performed to test the differences in fungal community composition throughout the three stages of alcoholic fermentation; a significant difference was observed between the two vineyard treatments ($F(1,16) = 18.1$, $R^2 = 0.53$, $p = 0.001$). *S. uvarum* dominated the early, mid, and late stages of alcoholic fermentation in both vineyard treatments. In the Vineyard 2 fermentations, *S. uvarum* made up 96.81 ± 0.2% of the relative abundance of these three stages, and *S. cerevisiae* made up 2.91 ± 0.2% (Fig 1A). In the Vineyard 8 fermentations, while *S. uvarum* still dominated with 82.88 ± 3.5% of the relative abundance and *S.*

*cerevisiae* was present at 0.89 ± 0.2%, a third yeast, *H. osmophila*, was also present, maintaining 14.56 ± 3.2% relative abundance through to the end of alcoholic fermentation (Fig 1B). Originally, it was thought that *Hanseniaspora* spp. could not survive during the later stages of alcoholic fermentation due to low ethanol tolerance, because these yeasts could not be isolated from later stages using culture-dependent methods [8, 90, 91]. However, *H. osmophila* has been shown to have an ethanol tolerance of at least 9% [92], and the advent of culture-independent identification techniques such as high-throughput amplicon sequencing have allowed for the identification of most microorganisms present in wine fermentations, including those in a viable but not culturable (VBNC) state. Indeed, other studies that have employed similar culture-independent techniques to ours have identified *Hanseniaspora* spp. through to the end of alcoholic fermentation [47, 71, 93–95]. *H. osmophila* has been characterized as a glucophilic yeast [92, 96, 97], and may have contributed to the alcoholic fermentation in the fermentations from Vineyard 8. However, it should be noted that the method of identifying the yeast community in this study involved analyzing DNA, not live cells, so it is possible that the DNA identified here as belonging to *H. osmophila* may not have come from living cells.

*S. uvarum* was also found dominating Chardonnay fermentations at this same winery two years previously [15], suggesting that this is a winery-resident population, capable of overwintering and entering fermentations year after year. Although it is possible that the *S. uvarum* population in this study was brought in from the vineyards with the grapes, we consider it more likely that the majority of these yeasts have established themselves as winery residents as opposed to vineyard residents. As mentioned above, *Saccharomyces* yeasts are typically rare on healthy grapes in the vineyard [40], and when *Saccharomyces* strains are identified in the vineyard, their presence is inconsistent between vintages [38, 98], and they are not necessarily the strains that conduct alcoholic fermentation in the winery [99, 100]. The presence of winery-resident yeast populations and communities is well-established in the literature [72, 91, 101–103], and these yeasts are capable of over-wintering in the winery and entering fermentations to which they have not been inoculated [4, 6, 7, 104]. *S. uvarum* is usually found in association with white wine fermentations in cool-climate wine regions. It has been identified dominating such fermentations in wineries across Europe, including France [12, 13], Hungary [14, 34], Italy [105], and Slovakia [14]. *S. uvarum* is known to be cryotolerant [18, 27], explaining its preference for low-temperature fermentations and cool-climate wine regions. The cellar in which the fermentations from this study were conducted is temperature-controlled and kept at 12˚C. This temperature is within a desirable growth range for *S. uvarum* [105, 106], but is too low for *S. cerevisiae* to grow optimally [34, 105]. Additionally, *S. uvarum* is a glucophilic yeast [33], and the residual fructose observed in the late stage of fermentation suggests that this yeast is less adept at fermenting fructose than *S. cerevisiae*. This result is supported by previous research [107].

To our knowledge, there is currently only one other winery (located in Alsace, France) that has been reported to have a local *S. uvarum* population dominating uninoculated fermentations across multiple vintages [12]. Interestingly, the Alsace cellar was also kept at 12˚C. It is possible that this temperature provides the optimal over-wintering conditions for *S. uvarum*, allowing it to out-compete *S. cerevisiae*, but more research is needed to investigate this.

## *Saccharomyces uvarum* strains

**Investigating the potential for *S. uvarum* hybrids.** Because both *S. cerevisiae* and *S. uvarum* were identified in these fermentations, we acknowledge the potential for *S. cerevisiae* x *S. uvarum* hybrids in this community. Originally, 47 yeast colonies were isolated for strain identification; they were plated on WLN media in order to differentiate between presumed *S.*

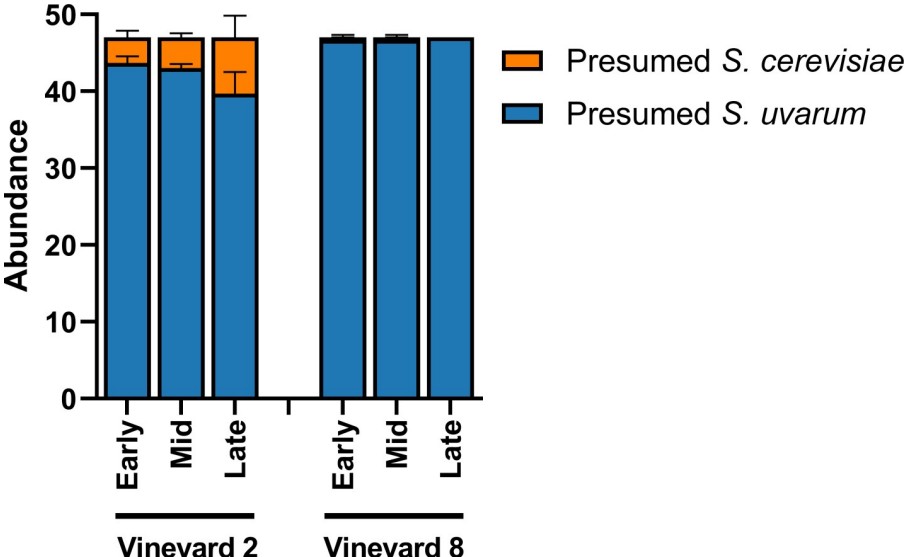

**Fig 2. Relative abundance of presumed *S. cerevisiae* and *S. uvarum*.** Relative abundance was determined based on phenotypical characteristics of presumed *S. cerevisiae* and presumed *S. uvarum* isolates present in three stages of fermentation (early, mid, and late) of Chardonnay sourced from two different vineyards (*n* = 3). Relative abundance was calculated from 47 yeast isolates per sample plated on Wallerstein Nutrient agar and presented ± SEM.

*cerevisiae* and presumed *S. uvarum* isolates (Fig 2). Additionally, in order to determine the potential colour of a hybrid strain on WLN plates, a pure *S. cerevisiae* strain (Lalvin BA11), a pure *S. uvarum* strain (CBS 7001), and a known *S. cerevisiae* x *S. uvarum* hybrid strain (Lalvin S6U) were all plated onto a WLN plate and incubated. The *S. cerevisiae* strain appeared as a cream-coloured colony, the *S. uvarum* strain appeared as a green colony, and the hybrid strain appeared as a cream-coloured colony (S2B Fig). Therefore, we conclude at least some hybrid strains may appear more similar to their *S. cerevisiae* parent when plated on WLN.

All but four of the presumed *S. cerevisiae* isolates were able to be strain-typed using primers that targeted *S. cerevisiae* microsatellite loci (for *S. cerevisiae* strain data see https://osf.io/j7rx8/ ). However, of the presumed *S. uvarum* isolates, a small proportion of the isolates only partially amplified when using primers that targeted *S. uvarum* microsatellite loci. These isolates were eventually removed from the analysis, and the *S. uvarum* population of each sample was rarefied to 32 *S. uvarum* strains, which represented the common denominator. The partial amplification of these isolates that were dropped from the analysis could suggest the presence of a small population of hybrid yeasts, as has been shown previously [41]. Another study conducted by Le Jeune et al. [108] characterized natural *S. cerevisiae* x *S. uvarum* hybrids where all loci amplified with primer sets from both species; however, only four *S. uvarum* microsatellite loci were targeted in that study.

To definitively determine if our *S. uvarum* strains were hybrids with *S. cerevisiae*, we conducted PCR-RFLP analysis on 50 isolates from this current study, representing the 50 most abundant strains identified. We amplified the ITS1 sequence of the rRNA gene, followed by digestion with the restriction enzyme *Hae*III, which results in DNA fragments of different sizes for *S. uvarum* (180 bp, 230 bp, and 500 bp) and *S. cerevisiae* (180 bp, 230 bp, and 320 bp). All 50 of our isolates matched the restriction pattern of the reference *S. uvarum* strain (CBS 7001) (S4 Fig). The reference *S. cerevisiae* strain (Fermol Mediterranée) had a different restriction pattern, while the reference hybrid strain (Lalvin S6U) had a pattern that combined the patterns of the two pure species (S4 Fig). Therefore, we are confident that the strains included

in the analysis of this study constitute pure *S. uvarum* strains, and not hybrids. Future studies specifically targeting *S. uvarum* hybrids will help determine if there is a significant hybrid population in the Okanagan wine region.

**S. uvarum strain diversity.**   *S. uvarum* isolates were strain-typed using 11 microsatellite loci. Prior to strain-typing, presumed *S. uvarum* colonies were distinguished from presumed *S. cerevisiae* colonies by plating them on WLN media (S2 Fig). Similar to the Illumina fungal community results above, we observed an increase in the relative abundance of presumed *S. cerevisiae* isolates in the fermentations from Vineyard 2 (Fig 2).

*S. uvarum* strain diversity was similarly high in the fermentations from both vineyards (Table 5), and this high diversity was maintained throughout alcoholic fermentation. No significant difference in strain diversity was observed between the two vineyards. The study conducted in 2015 at this same winery also reported high *S. uvarum* strain diversity that was established early and was maintained throughout alcoholic fermentation [15]. This result has also been observed with regards to *S. cerevisiae* strain diversity in uninoculated commercial fermentations [6, 109].

## S. uvarum strain composition

A total of 106 unique *S. uvarum* strains were identified across the six fermentations sampled in this study (S3 Table). Two years previously, 150 unique *S. uvarum* strains were identified in fermentations at this same winery [15]; 66 of these strains (44–62% of all strains) were identified in both vintages. The previous study strain-typed 1,860 yeast isolates, in comparison to the 576 isolates strain-typed in this current study, so it is expected that fewer strains would be identified here. This large overlap in strains identified across vintages further suggests that most of these strains are winery residents: previous research investigating vineyard-derived *S. cerevisiae* strains found only a 9–13% overlap in successive vintages [38]. Other studies conducted to investigate *S. uvarum* strain abundance have identified far fewer strains (a maximum of 89 unique strains isolated from grapes, wine, and other environments), although it should be noted that fewer isolates (up to 114) were strain-typed in those studies, potentially explaining this discrepancy [12–14, 22, 28, 41, 105, 110].

Of the 106 strains identified in this study, only four were able to achieve $\geq$ 10% relative abundance in at least two samples (Fig 3). The other 102 strains, termed 'minor strains,' accounted for ~ 50% of the relative abundance in the Vineyard 2 fermentations, and ~ 65% of the relative abundance in the Vineyard 8 fermentations. The two most abundant strains in both vineyard treatments ('2015 Strain 2' and '2015 Strain 3') were first identified as dominant strains at the same winery two years previously, during the 2015 vintage [15]. The reoccurrence of '2015 Strain 2' and '2015 Strain 3' after two years suggests that these strains have established themselves as persistent winery residents, capable of entering and dominating fermentations in multiple vintages. The other two dominant strains identified in these

**Table 5.  *Saccharomyces uvarum* strain diversity, measured as Simpson's Index of Diversity (1 − D) and Shannon's Diversity Index (H), of stainless steel barrel-fermented Chardonnay sourced from two different vineyards.**  Diversity ± SEM was measured from three stages of fermentation in the winery: early, mid, and late (*n* = 3). For each diversity index, a one-way repeated-measures ANOVA was performed to compare the differences between vineyards across all fermentation stages. The *p*-values for each index are indicated in the appropriate columns, and any significant differences ($p \leq 0.05$) are in bold.

| | Simpson's Index (1−D) | | | Shannon's Index (H) | | |
|---|---|---|---|---|---|---|
| Sample | Vineyard 2 | Vineyard 8 | *p* = | Vineyard 2 | Vineyard 8 | *p* = |
| Early | 0.88 ± 0.02 | 0.92 ± 0.009 | 0.19 | 2.46 ± 0.16 | 2.75 ± 0.08 | 0.09 |
| Mid | 0.89 ± 0.01 | 0.90 ± 0.001 | | 2.51 ± 0.04 | 2.59 ± 0.07 | |
| Late | 0.85 ± 0.03 | 0.88 ± 0.04 | | 2.30 ± 0.23 | 2.57 ± 0.26 | |

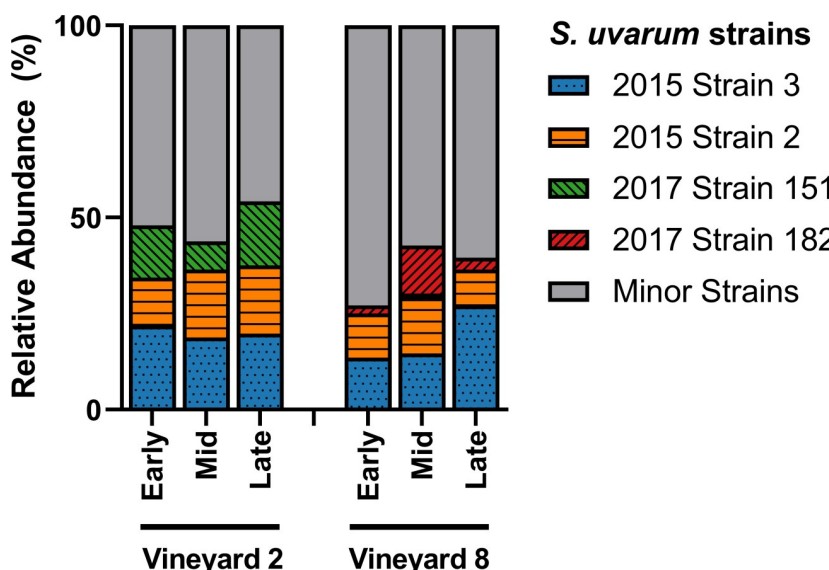

**Fig 3. *S. uvarum* strain abundance.** Relative abundance of the dominant *S. uvarum* strains present in three stages of fermentation (early, mid, and late) of Chardonnay sourced from two different vineyards ($n$ = 3). Relative abundance was calculated from 32 *S. uvarum* yeast isolates per sample. Any strains that did not achieve ≥ 10% relative abundance in at least two samples were grouped into the Minor Strains category. For variation among samples please see S4 Table.

fermentations ('2017 Strain 151' and '2017 Strain 182') were not isolated during the 2015 vintage, and were also not evenly distributed among the fermentations from the two vineyards (Fig 3). The '2017 Strain 151' represented ~ 7–17% of the relative abundance in the Vineyard 2 fermentations, and was only identified at ~ 1% in a single sample in the Vineyard 8 fermentations. Meanwhile, the '2017 Strain 182' represented ~ 2–12% of the relative abundance in the Vineyard 8 fermentations, and was not identified at all in the Vineyard 2 fermentations. This difference in dominant strains could be attributed to a number of causes, including differences in acidity levels in the musts from the two different vineyards, or simply changes in the winery environment over time, since these fermentations did occur approximately two weeks apart. A significant correlation between grape must acidity and the dominance of specific *S. cerevisiae* strains has been previously observed [111], so it is understandable that a similar result might be expected with regards to *S. uvarum* strains. It is also possible that these two differentially-abundant strains are predominantly vineyard-residents, with '2017 Strain 151' present in (or originating from) Vineyard 2, and '2017 Strain 182' present in (or originating from) Vineyard 8; this could explain their increased presence in one vineyard treatment over the other, but more research is needed in order to determine if this is the case.

A PERMANOVA was performed to test the differences in *S. uvarum* population composition throughout the three stages of alcoholic fermentation; a significant difference was observed between the two vineyard treatments ($F(1,16)$ = 3.7, $R^2$ = 0.19, $p$ = 0.001). A principal coordinates analysis (PCoA) was also generated in order to visualize the differences in *S. uvarum* strain composition between the fermentations from the two vineyards while including all 106 strains (Fig 4). The PCoA ordination showed a clear separation between samples taken from the two vineyards. This result highlights the importance of analyzing not only the diversity of a sample but also the composition. In this study, as in previous studies of a similar nature [15, 47], treatments have been found to have near-identical diversity values but completely distinct compositions. This is because composition considers not only the relative abundance of different strains, but also the identities of those strains, and can provide a more

accurate summary of the differences observed among treatments. Previous research has identified that different strains of *S. uvarum* can produce wines with different sensory-active metabolite profiles, especially when fermented at lower temperatures [32, 33], and it is therefore expected that different *S. uvarum* populations would contribute differently to the production of wine sensory attributes. However, this is still a new topic of investigation, and more research is needed in this area to determine whether the differences in secondary metabolite composition among wines fermented with different *S. uvarum* strains translate into detectable differences in the sensory profiles of the wines.

## *S. uvarum* genetic diversity

Of the 106 strains identified in this study, 66 were also identified in the 2015 vintage at the same winery, including all four of the dominant strains from the 2015 vintage [15]. Additionally, we noted that 32 strains were unique to the Vineyard 2 fermentations, 40 strains were unique to the Vineyard 8 fermentations, and 34 strains were found in both the Vineyard 2 and Vineyard 8 fermentations. Although some minor strains were found to be unique to specific stages of fermentation, these strains were found in very low abundance: of the top 20 strains with the highest overall abundance, 18 were identified in all three stages of fermentation, suggesting that ethanol tolerance is not a major contributor to any differences in strain abundance observed here.

Four minor strains were found to bear genetic similarity to international strains (within a Bruvo distance of 0.3). Seven isolates from the Vineyard 2 fermentations were genetically similar to the commercial strain Velluto BMV58®, despite this strain not being sold in Canada at the time of this study. Additionally, three isolates were genetically similar to the previously-sequenced Spanish strain CBS 7001, one isolate was genetically similar to the French strain CBS 8711, and one isolate was genetically similar to strain PYCC 6860, which was isolated from oak trees on Hornby Island, in the same province of Canada as the winery from this study (although separated by hundreds of km). Local strains bearing genetic relatedness to international strains has been previously observed; some *S. uvarum* strains isolated in New Zealand were also found to bear significant nucleotide similarity with CBS 7001 [28].

The probability that two unrelated individuals would have identical multilocus genotypes (MLGs) was calculated to be $P_I = 6.7 \times 10^{-8}$. However, this probability does not take into account the Bruvo distance used in this study; individuals in this study could have slightly different MLGs and still be grouped into the same strain classification. Fixation indices were also calculated for this *S. uvarum* population, comparing the subpopulation identified in the Vineyard 2 fermentations with the subpopulation identified in the Vineyard 8 fermentations. The proportion of genetic variance contained within each vineyard treatment subpopulation relative to the entire population ($F_{ST}$) was calculated to be 0.014 ± 0.003, indicating that the two subpopulations share a high degree of genetic material and suggesting a high level of interpopulation breeding. The inbreeding coefficient ($F_{IS}$) was calculated to be 0.74 ± 0.04, indicating a considerable degree of inbreeding.

The occurrence of heterozygosity in this study was higher than was observed at the same winery two years previously: 51.9% of the strains in this study contained at least one heterozygous locus, as compared to 42.7% in 2015 [15]. This is the highest incidence of heterozygosity observed in an *S. uvarum* population to-date: previous studies have found heterozygous loci in 28.8% [41], 23.1% [28], and 0% [22] of *S. uvarum* strains isolated from grapes and wine. However, these studies did contain fewer isolates and analyzed fewer hypervariable microsatellite loci, which may explain some of the differences observed. The L9 microsatellite locus was the most variable by far, containing 18 different alleles in this study. The second most variable

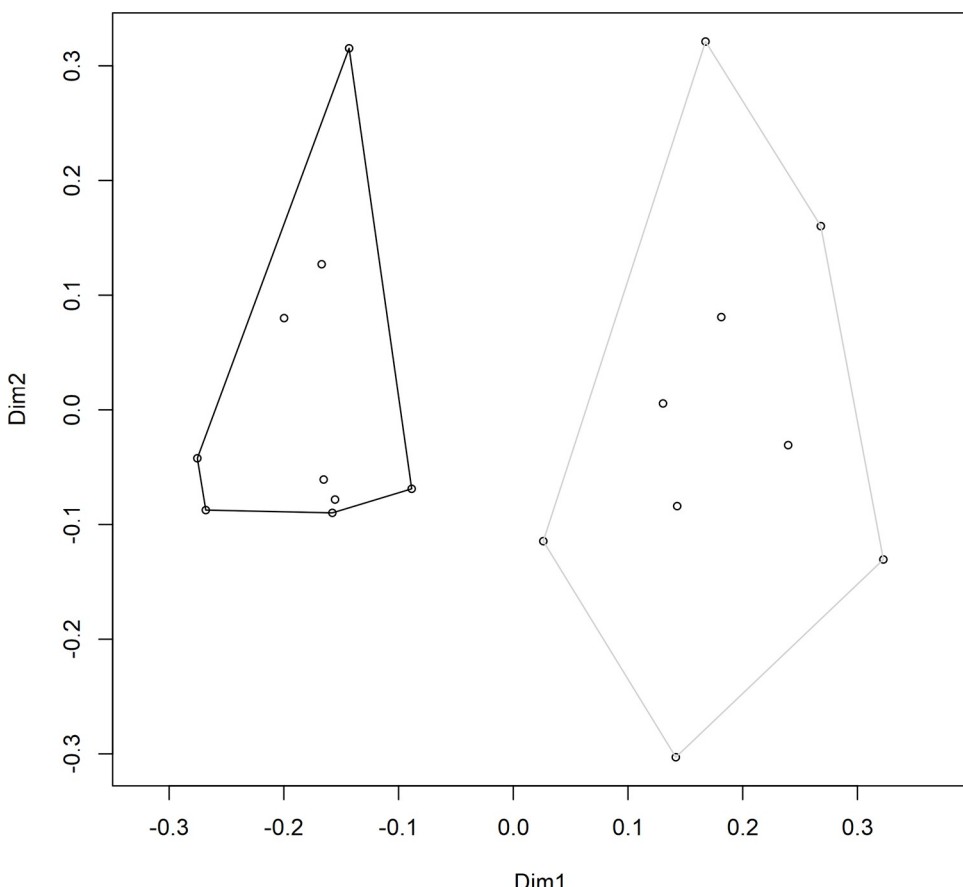

**Fig 4. *S. uvarum* strain composition.** Principal coordinates analysis (PCoA) ordination of the *S. uvarum* strain composition in Chardonnay wines sourced from two vineyards: Vineyard 2 (black) and Vineyard 8 (grey). Individual data points represent the composition of *S. uvarum* strains in a single sample (based on 32 yeast isolates per sample). Samples were taken at three stages of alcoholic fermentation, and each vineyard treatment contained three biological replicates, for a total of nine samples per vineyard. Dimension 1 (Dim1) explains 20.98% of variance, and Dimension 2 (Dim2) explains 13.07% of variance.

locus was L2, containing eight different alleles. The other loci contained either five alleles (L7 and L8), four alleles (L1, L3, L4, and NB9), or three alleles (NB1, NB4, and NB8). While the rate of heterozygosity was higher in this population than has been previously observed, the observed heterozygosity rate for each locus was between 2 and 16 times lower than the unbiased expected heterozygosity rate for each locus, suggesting that this population still has a high selfing rate (for supporting data, visit https://osf.io/j7rx8/).

Population structure was assessed by performing Bayesian clustering on the 102 indigenous *S. uvarum* strains identified in the 2017 winery fermentations, as compared to 12 international strains (Table 2), using InStruct [62]. This method, which assigns strains membership coefficients to different inferred ancestors and takes into account inbreeding rates, has been previously used to assess the structure of both *S. cerevisiae* [9, 38, 112, 113] and *S. uvarum* [41] populations. Both K = 11 and K = 5 clusters were identified depending on the analysis method, and both were plotted to visualize the inferred ancestry of each strain using DISTRUCT (Fig 5).

A dominant inferred ancestor was identified if the inferred coefficient was equal to or higher than 0.75 (75%) of the total ancestry; a similar metric has been used previously to infer dominant ancestry in *S. uvarum* populations [41]. For the K = 11 analysis (Fig 5B), no strain had

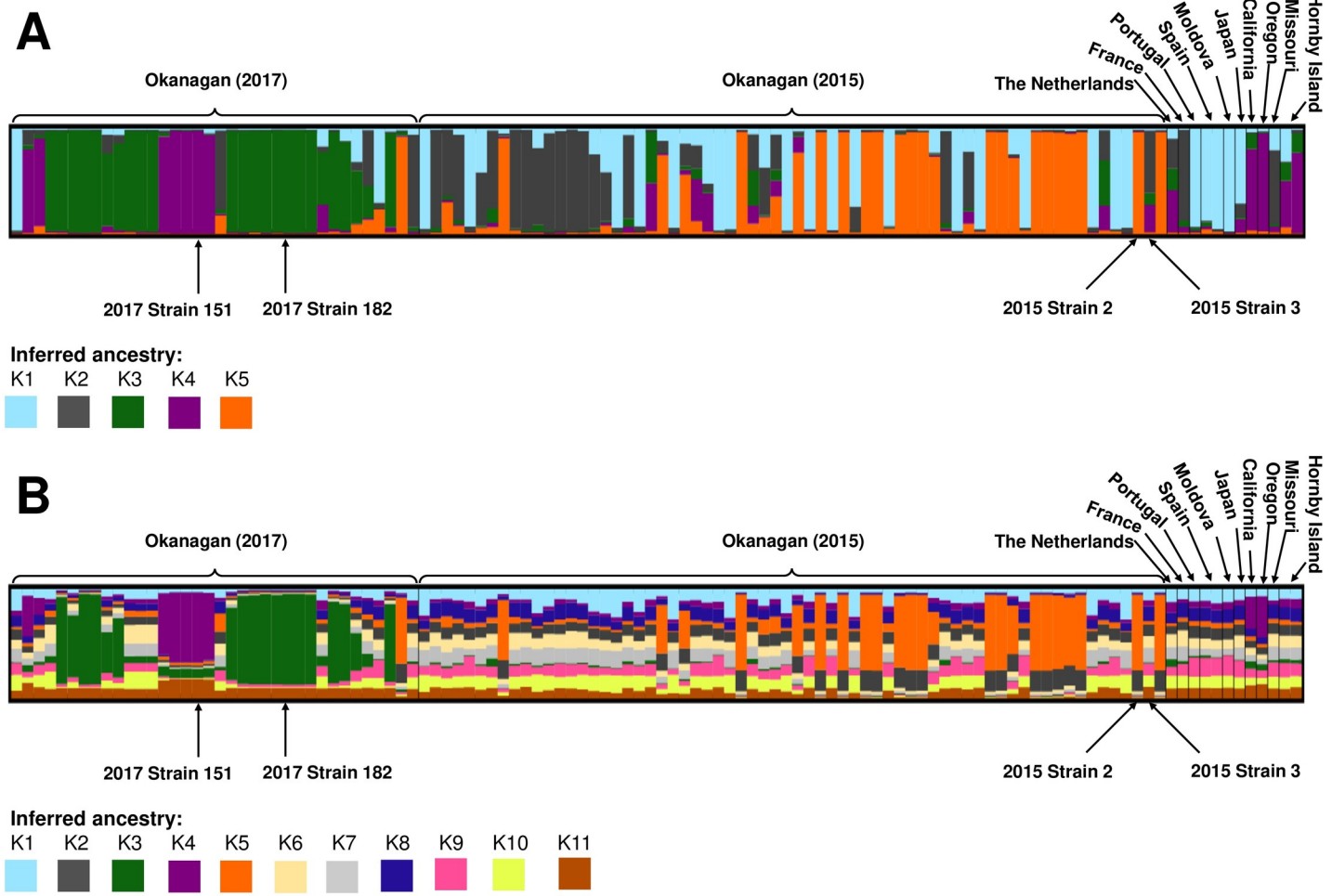

**Fig 5. DISTRUCT plots of *S. uvarum* population structure by inferred ancestry.** Each column represents a single *S. uvarum* strain, and different colours correspond to different inferred ancestors: (A) K1 to K5 inferred ancestry clusters, and (B) K1 to K11 inferred ancestry clusters. Strains included in this analysis include 12 international strains, as well as 102 indigenous strains isolated at an Okanagan winery during the 2017 vintage. Strains in the Okanagan (2015) section were originally identified at the same winery during the 2015 vintage, and were isolated again in 2017. Strains in the Okanagan (2017) section were unique to the 2017 vintage. The four dominant strains identified in this study are indicated.

a single ancestor, and most strains were not dominated by any one subpopulation; instead, the strains appeared to have several ancestors, suggesting significant admixture between subpopulations. For the K = 5 analysis, most of the strains contained one dominant ancestor, although all strains contained some contribution from multiple inferred ancestors (Fig 5A). Many of the Okanagan strains identified in this study had a dominant inferred ancestor that was distinct from the international strains. Furthermore, there was a stark distinction in inferred ancestry profiles between the strains originally identified in 2015 and those identified for the first time in 2017, suggesting a strong vintage-to-vintage variability in at least a portion of the *S. uvarum* population. Indeed, an analysis using ObStruct [66] found that both the geographic origin and the vintage of origin (in the case of the Okanagan strains identified in this study) influenced population structure ($R^2 = 0.28$, $p < 0.001$); interestingly the output of the ObStruct analysis also suggested that the predefined population of Okanagan *S. uvarum* strains identified for the first time in 2017 was the greatest driver of population structure (https://osf.io/j7rx8/).

Twenty of the strains originally identified in 2015 had K5 (orange) as the dominant inferred ancestor; none of the international strains had this structure, and only one of the 2017 strains

had K5 (orange) as their dominant inferred ancestor (Fig 5A). Additionally, 10 of the 2015 strains had K2 (grey) as the dominant inferred ancestor, which was the case for just one 2017 strain and one international strain (CBS 8711 from France). For the strains unique to the 2017 vintage, more than half had K3 (green) as the dominant inferred ancestor; no international strains nor any 2015 strains had K3 (green) as a dominant inferred ancestor, making this result unique and specific to the strains from that vintage. Indeed, K3 (green) comprises on average less than 5% of the ancestry profiles of the international and 2015 strains. Additionally, seven of the 2017 strains had K4 (purple) as a dominant inferred ancestor. While none of the 2015 strains had this result, three international strains also had K4 (purple) as their dominant inferred ancestor, all from the west coast of North America: CBS 8696 (California), PYCC 6901 (Oregon), and PYCC 6861 (Hornby Island, British Columbia, Canada). Finally, a number of strains, both of international original and from this study, had K1 (blue) as the dominant inferred ancestor: these included one strain from 2017, 17 strains from 2015, and four international strains, from Portugal (PYCC 6871), Spain (CBS 7001 and BMV58), and Moldova (CBS 8690). The four dominant strains from this study all had different inferred ancestry. '2017 Strain 151' had K4 (purple) as its dominant inferred ancestor, '2017 Strain 182' had K3 (green) as its dominant inferred ancestor, and '2015 Strain 2' had K5 (orange) as its dominant inferred ancestor; for each of these strains, the dominant inferred ancestor represented almost 100% of the ancestry profile (coefficients 0.979, 0.976, and 0.971, respectively). Contrastingly, '2015 Strain 3' did not contain a dominant inferred ancestor, and its inferred ancestry profile was split between K2 (grey), K3 (green), and K4 (purple) (coefficients 0.596, 0.110, and 0.260, respectively). The strain with the closest inferred ancestry profile to '2015 Strain 3' was strain CBS 395 from The Netherlands (coefficients K2 = 0.284, K3 = 0.205, K4 = 0.402.

The dominant strains from this study had a similar structure using both the K = 5 and the K = 11 clustering methods. In both cases, '2017 Strain 151' had K = 4 (purple) as a majority ancestor, '2017 Strain 182' had a K = 3 (green) as a majority ancestor, and '2015 Strain 2' had K = 5 (orange) as a majority ancestor, and '2015 Strain 3' comprised multiple inferred ancestors, with none dominating.

A number of the Okanagan strains identified in this study (including one of the dominant strains), along with the west coast international strains from California, Oregon, and Hornby Island, contained K4 (purple) as a dominant or significant inferred ancestor. This suggests that these strains may have a shared geographical origin. Aside from these west coast strains, most of the Okanagan strains from both vintages had different majority inferred ancestors from the international strains. In particular, 41 of the 102 Okanagan strains had a K3 (green) or K5 (orange) dominant inferred ancestor, neither of which appeared in significant proportions in the inferred ancestry profiles of the international strains (Fig 5A). This demonstrates that a significant majority proportion of the Okanagan strains have unique inferred ancestry not seen elsewhere in the world, further bolstering the idea of the presence of Okanagan-specific *S. uvarum* population.

A phylogenetic tree was also created in order to visualize the genetic relatedness of the 102 indigenous Okanagan *S. uvarum* strains compared to the 12 international *S. uvarum* strains (Fig 6). These strains were differentiated into six sub-populations using the same K = 5 clustering output from InStruct that was used to visualize population structure in the DISTRUCT plots, which used Bayesian clustering to find inferred ancestry of the strains. Strains were coloured based on dominant inferred ancestry clustering, defined as representing at least 75% of the inferred ancestry profile based on the InStruct analysis for K = 5 clusters. Strains without a dominant inferred ancestor were left uncoloured (black). This clustering method identified two ancestors that appear unique to the Okanagan, and to each vintage: K3 (green) is unique to 2017, while K5 (orange) is unique to 2015. Additionally, the inferred ancestor K4 (purple)

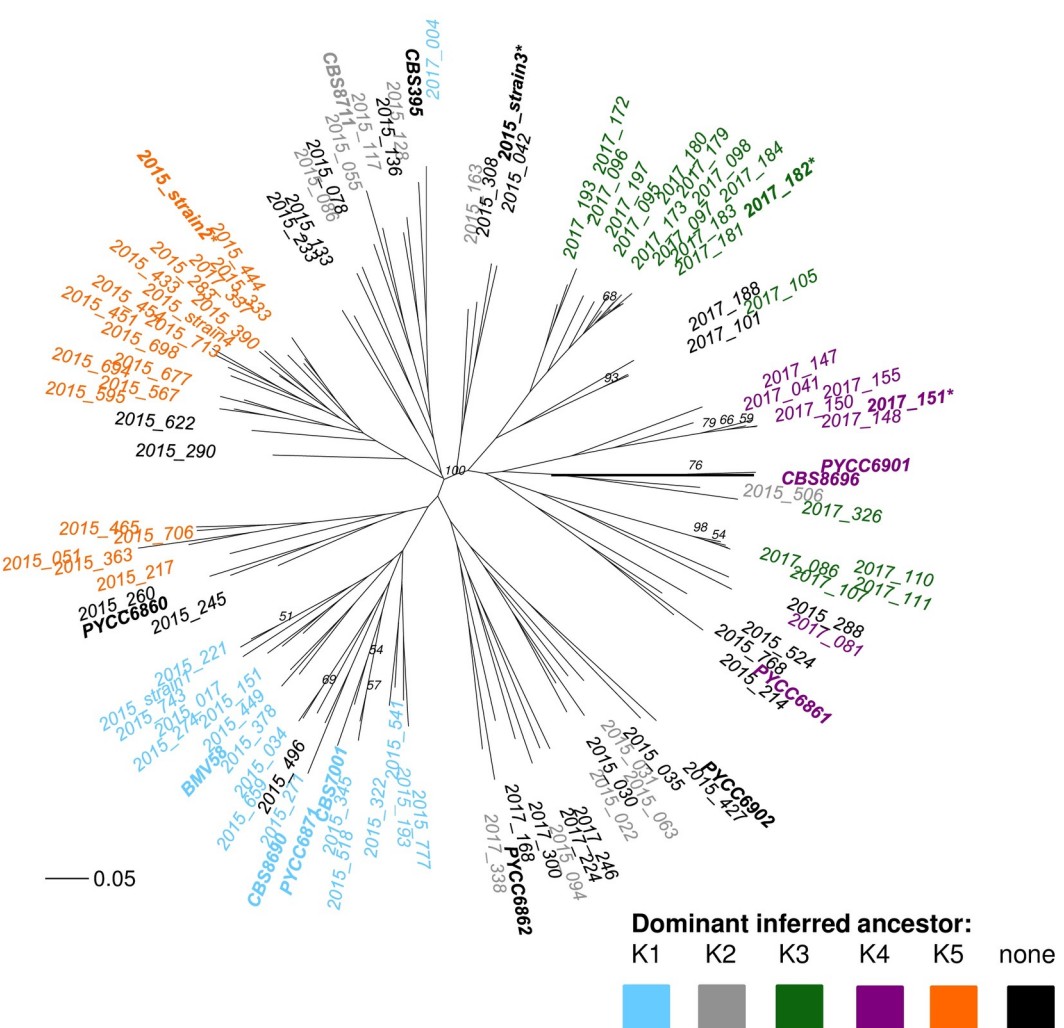

**Fig 6. Phylogenetic tree of Okanagan and international *S. uvarum* strains.** An unrooted, neighbour-joining phylogenetic tree using Bruvo distance, comparing the genetic relatedness of the *S. uvarum* strains identified in Chardonnay fermentations in the Okanagan Valley (Canada) during the 2017 vintage, as well as 12 selected *S. uvarum* strains isolated from around the world (see Table 2 for strain origins). The international strains are shown in bold, and the four dominant strains identified in this study are shown with an asterisk. Strains are coloured based on dominant inferred ancestry clustering, defined as representing at least 75% of the inferred ancestry profile based on an InStruct analysis for K = 5 clusters. Strains without a dominant inferred ancestor were left uncoloured. A Bruvo distance of 0.05 is shown for scale. Bootstrap values above 0.5 are indicated.

was identified as a Pacific west coast ancestor, because it contained Okanagan strains as well as strains from other Pacific west coast locations. The strains with K5 (orange) as their dominant inferred ancestor grouped together on the phylogenetic tree (Fig 6). Interestingly, the strains with K3 (green) or K4 (purple) as their dominant inferred ancestor grouped closely together, suggesting that these strains (most of which were identified in 2017) are genetically similar, and that these two inferred ancestors may have been closely related. The strains with K1 (blue), as their dominant inferred ancestor clustered together, separate from the other strains. This group contained Okanagan strains first identified during the 2015 vintage as well as international strains from Europe (Spain, Portugal, and Moldova). Finally, the strains with K2 (grey) as their dominant inferred ancestor grouped closely with the strains that had no dominant ancestor, and were spread over multiple branches of the phylogenetic tree. These strains

had K2 (grey) as a dominant ancestor only in the K = 5 clustering method (Fig 5): in the K = 11 clustering method, these strains all contained multiple minor inferred ancestors (none of them dominant), which can explain why they are seen clustering with the strains that have no dominant inferred ancestry in this tree. The four dominant strains from this study (in bold and marked with an asterisk in Fig 6) all belonged to separate clusters, as was expected based on their inferred ancestry profiles (Fig 5).

The only commercial *S. uvarum* strain included in the construction of our phylogenetic tree was Velluto® BMV58 (Lallemand, Montreal, QC, Canada). Recently, one additional *S. uvarum* strain and one hybrid *S. uvarum* × *S. cerevisiae* strain have been commercially released: VitiFerm™ Sauvage BIO and EnartisFerm® ES U42, respectively. Some enological properties of *S. uvarum* have been studied, but more research is needed to allow winemakers to make informed decisions when selecting these non-traditional yeast strains for inoculation. *S. uvarum* produces lower levels of ethanol, acetic acid, and acetaldehyde, and higher levels of glycerol, succinic acid, malic acid, isoamyl alcohol, isobutanol, and ethyl acetate, as compared to *S. cerevisiae* [31–34]. Furthermore, due to its relatively lower production of ethanol, *S. uvarum* has been suggested as a potential means of mitigating the effects of climate change on winemaking [114]. With a warming climate, many winemaking regions are beginning to produce very ripe grapes with high sugar contents, which, with traditional fermentation techniques, can result in wines with very high ethanol contents. If this trend continues, many wines produced in the future could contain alcohol concentrations above the legal regulations of some countries, since diluting grape must with water is not permitted in wine production. Using non-traditional yeasts such as *S. uvarum* or non-*Saccharomyces* species, which can metabolize grape sugars into compounds other than ethanol (such as glycerol), may help mitigate this issue by keeping ethanol production within permitted levels. The recent increase in the availability of commercial *S. uvarum* strains indicates that this is clearly an area of interest for winemakers, and our changing climate and consumer preferences require investigation into new and creative methods of wine production, including the use of non-traditional yeasts such as *S. uvarum*, either alone or in combination with *S. cerevisiae*.

## Conclusions

This study investigated the fungal communities and *S. uvarum* populations present in uninoculated commercial Chardonnay fermentations of grapes that originated from two different vineyards. Differences in fungal community composition were observed, with *H. osmophila* representing a larger proportion of the fungal community in the fermentations from one vineyard over the other. However, in all of the fermentations, *S. uvarum* was the dominant yeast during the early, mid, and late stages of alcoholic fermentation. An investigation into the genetic diversity of the *S. uvarum* strains present in this study was conducted, and this population was found to be both highly diverse and genetically distinct from *S. uvarum* strains identified in other regions of the world. A total of 106 *S. uvarum* strains were identified in this study, but only four strains played a dominant role in fermentation; two of these dominant strains were also identified as dominant strains at this same winery two years previously. The presence of persistent non-commercial strains, as well as the population structure analysis generated to compare Okanagan and international strains, provides evidence for an indigenous *S. uvarum* population with unique genotypes in the Okanagan Valley of British Columbia, Canada.

## Supporting information

**S1 Fig. Vineyard sampling map.** Sampling layouts for (A) Vineyard 2 and (B) Vineyard 8. Each of the two conjoined squares represent the generated site of collection for one sample,

with the sample number also given at each sampling site. One sampling site contains approximately 15 vines, and two grape clusters were taken from each vine (one on either side of the row), for a total of 30 clusters per sampling site. The geographic orientation of each vineyard is indicated in the bottom right corner of each sampling map.
(DOCX)

**S2 Fig. Yeast isolate plating on Wallerstein Nutrient (WLN) media.** (A) Forty-seven yeast colonies per barrel, per sampling stage, were isolated and plated onto WLN media in order to distinguish between presumed *S. cerevisiae* and presumed *S. uvarum* isolates prior to strain-typing. On each plate, an *S. cerevisiae* control (Lalvin BA11) and an *S. uvarum* control (CBS 7001) were used to help aid the differentiation between the two species. Presumed *S. cerevisiae* isolates appeared cream-coloured, while presumed *S. uvarum* isolates appeared green. (B) Comparison of colony colour of a pure *S. cerevisiae* strain (Lalvin BA11), a pure *S. uvarum* strain (CBS 7001), and a *S. cerevisiae* x *S. uvarum* hybrid (Lalvin S6U), plated on WLN media.
(DOCX)

**S3 Fig. Histogram of pairwise genetic distances of multilocus genotypes analyzed in poppr (R package).** The plot shows genetic distance cutoff as a function of the number of multilocus lineages, depending on the clustering method used.
(DOCX)

**S4 Fig. Restriction analysis of yeast DNA using *Hae*III.** Restriction digest profiles of 50 yeast isolates, representing the 50 most abundant strains found in this study, as compared to reference strains: a pure *S. cerevisiae* strain (Fermol Mediterranée), a pure *S. uvarum* strain (CBS 7001), and a *S. cerevisiae* x *S. uvarum* hybrid strain (Lalvin S6U). (A) First 25 strains, as well as a ladder and all three reference strains. (B) Second 25 strains, as well as a ladder and all three reference strains.
(DOCX)

**S5 Fig. Rarefaction curves.** Rarefaction curves featuring (A) species richness in the fungal community, and (B) strain richness in the *S. uvarum* community, at different sampling depths.
(DOCX)

**S1 Table. Results of Levene's test for equal variance among groups for all chemical parameters measured.** Any significant differences ($p \leq 0.05$) are in bold.
(DOCX)

**S2 Table. Fungal community composition of Chardonnay grapes, must, and fermenting wine sourced from two different vineyards, based on 20,000 sequences per sample and represented as percent (%) relative abundance.** Samples were taken from grapes in the vineyard (G), and at four stages of fermentation in the winey: cold settling (C), early (E), mid (M), and late (L). Vineyard 2 grape sample values are the means ± SEM of five replicates, and Vineyard 8 grape sample values are the means ± SEM of 6 replicates. All winery fermentation stages have three reported replicates, with the exception of the cold settling stage from the Vineyard 8 fermentations, which contained two. Sequences were identified to the species level unless otherwise indicated. Fungal species that represented less than 10% of the relative abundance in at least two samples were grouped into the Minor Fungi category. One exception was *Saccharomyces cerevisiae*, which never reached 10% relative abundance in any sample but is included in this table because of its importance during alcoholic fermentation. The last two columns indicate positive (Pos) and negative (Neg) controls. For the raw data containing all the fungi identified in this study (including minor fungi), please visit https://osf.io/j7rx8/.
(DOCX)

**S3 Table. Microsatellite identities of the representative multilocus genotypes (MLGs) of *Saccharomyces uvarum* strains isolated from stainless steel barrel-fermented Chardonnay at Canadian winery during the 2017 vintage.** Allele sizes for allele 1 (A1) and allele 2 (A2) are shown for each of the 11 microsatellite loci analyzed. Strains with the prefix "2017" were isolated exclusively during the 2017 vintage. Strains with the prefix "2015" were previously isolated and characterized during the 2015 vintage at the same winery, and were also isolated during the 2017 vintage. Strains without a vintage prefix are those that belong to global yeast databases (see S5 Table).
(DOCX)

**S4 Table. *Saccharomyces uvarum* population composition of Chardonnay grapes, must, and fermenting wine sourced from two different vineyards, based on 32 yeast isolates per sample.** Samples were taken at three stages of fermentation: early (E), mid (M), and late (L). Values are the means ± SEM ($n$ = 3). Strains that represented less than 10% of the relative abundance in at least two samples were grouped into the Minor Strains category. For the raw data, containing all the *S. uvarum* strains identified in this study (including minor strains), please visit https://osf.io/j7rx8/.
(DOCX)

**S5 Table. Microsatellite identities of the multilocus genotypes (MLGs) of *Saccharomyces uvarum* strains obtained from global yeast databases.** Allele sizes for allele 1 (A1) and allele 2 (A2) are shown for each of the 11 microsatellite loci.
(DOCX)

## Acknowledgments

The authors would like to thank the winemakers Darryl Brooker, Corrie Krehbiel, and Alexandra Haselich of Mission Hill Family Estate Winery for their assistance, guidance, and donation of fermentation samples. We also thank Britney Johnston and Mehrbod Estaki of the University of British Columbia for technical support and high-throughput amplicon sequencing data analysis support, respectively.

## Author Contributions

**Conceptualization:** Garrett C. McCarthy, Jonathan T. Martiniuk, Vivien Measday, Daniel M. Durall.

**Data curation:** Garrett C. McCarthy, Sydney C. Morgan, Jonathan T. Martiniuk, Brianne L. Newman.

**Formal analysis:** Garrett C. McCarthy, Sydney C. Morgan.

**Funding acquisition:** Vivien Measday, Daniel M. Durall.

**Investigation:** Garrett C. McCarthy, Sydney C. Morgan, Brianne L. Newman, Stephanie E. McCann.

**Methodology:** Garrett C. McCarthy, Sydney C. Morgan, Jonathan T. Martiniuk, Brianne L. Newman, Stephanie E. McCann.

**Project administration:** Vivien Measday, Daniel M. Durall.

**Software:** Sydney C. Morgan.

**Supervision:** Vivien Measday, Daniel M. Durall.

**Validation:** Garrett C. McCarthy, Sydney C. Morgan.

**Visualization:** Garrett C. McCarthy, Sydney C. Morgan.

**Writing – original draft:** Garrett C. McCarthy, Sydney C. Morgan.

**Writing – review & editing:** Garrett C. McCarthy, Sydney C. Morgan, Jonathan T. Martiniuk, Brianne L. Newman, Stephanie E. McCann, Vivien Measday, Daniel M. Durall.

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
