## [Decision Letter · Decision Letter 0]

28 Apr 2020

PONE-D-19-30860

An indigenous Saccharomyces uvarum population with high genetic diversity dominates uninoculated Chardonnay fermentations at a Canadian winery

PLOS ONE

Dear Dr. Morgan,

Thank you for submitting your manuscript to PLOS ONE. After careful consideration, we feel that it has merit but does not fully meet PLOS ONE’s publication criteria as it currently stands. Therefore, we invite you to submit a revised version of the manuscript that addresses the points raised during the review process, in particular the following major points raised by referees: 1- S. cerevisiae x S. uvarum hybrid detection, 2- the use of another software than adegenet for the detection of admixture, 3- statistical issues raised by second referee, such as Kruskal-Wallis text vs ANOVA, and statistical evaluation of communities, 4- taking into account heterozygosity at microsatellite loci. All minor points raised by referees should be addressed as well. 

This will greatly improve the manuscript, and help publish your work that has been well appreciated by reviewers.

We would appreciate receiving your revised manuscript by Jun 12 2020 11:59PM. To enhance the reproducibility of your results, we recommend that if applicable you deposit your laboratory protocols in protocols.io, where a protocol can be assigned its own identifier (DOI) such that it can be cited independently in the future. For instructions see: http://journals.plos.org/plosone/s/submission-guidelines#loc-laboratory-protocols

We look forward to receiving your revised manuscript.

Kind regards,

Cecile Fairhead, Ph.D.

Academic Editor

PLOS ONE

Journal Requirements:

Additional Editor Comments (if provided):

Reviewers' comments:

Reviewer's Responses to Questions

**Comments to the Author**

1. Is the manuscript technically sound, and do the data support the conclusions?

Reviewer #1: Yes

Reviewer #2: No

2. Has the statistical analysis been performed appropriately and rigorously? 

Reviewer #1: Yes

Reviewer #2: No

3. Have the authors made all data underlying the findings in their manuscript fully available?

Reviewer #1: Yes

Reviewer #2: Yes

4. Is the manuscript presented in an intelligible fashion and written in standard English?

Reviewer #1: Yes

Reviewer #2: Yes

5. Review Comments to the Author

Reviewer #1: In this manuscript McCarthy et al. present the characterization of a unique yeast population from a Canadian winery. Interestingly, this winery is inhabited by a S. uvarum population, which makes it one of the few example of yeast microflora not containing Saccharomyces cerevisiae. It presents a comprehensive analysis of the fungal population from the vineyard to the vats, and an analysis of S. uvarum population in the winery and the vineyard.

This original, well led and well written study is an interesting case study of alternative micro-flora under cold climate. This case was known for cool climate vineyards, but until now, this is the first study presenting this situation.

Despite the quality of the manuscript, I feel one main lack and few missing piece of information or analyses.

1 – The authors have selected theirs strains on Wallerstein nutrient media. This is not a robust species identification method, and authors have not explored for the presence of S. cerevisiae x S. uvarum interspecic hybrids that may participate to the ecosystem. Doing so they may have omitted one component of the fermentation microflore. This had been described in one of the first winery containing S. uvarum strains (Demuyter et al. 2014 J. Applied Microbiol , LeJeune et al. 2007 FEMS Yeast Research), and I personally have detected many others in cold cellars. As a consequence, it is necessary to check for the presence of hybrids among the “S. uvarum” isolates (that microsatellite considered as S. uvarum) or among the S. cerevisiae. An overview of the frequencies of the counts of S. uvarum and S. cerevisiae in the different samples is missing

2 – When analyzing their population structure, with the package adegenet, the authors mention that the topology of the tree was not matching the clustering obtained with the “find.cluster function of the adegenet package. This function is very efficient for the identification of groups, but less for the identification of admixed populations. Alternative populations structure software enable such inferences : i.e. InStruct, or Structure softwares take into account admixture, making the assumption of an inbred population (described as the case in manuscripts of other groups) or maximizing the likelihood of Hardy Weinberg equilibrium.

3 – Some minors point

Line 127-147 : what is the total weight of the grape samples?

Line 364: the distance used for the estimation of dissimilarity is not indicated: the function “vegdist” of the vegan package uses several dissimilarity distances (i.e : Bray-Curtis, Jaccard…)

Line 424 : the unit in which volatile acidity is expressed is not indiccated : is this g/l of acetic acid, or H2SO4? This threshold is very high and caused by the choice of the analytical method. Other methods (i.e. enzymatic kits, or distillation) provide much higher sensitivity.

Line 439-449: Table 4 : Why only Simpson’s Index is given. Other indices such as the popular Shannon index provide complementary information on the diversity isolated in the samples. A rarefaction method should be used in order to avoid variations of sample size (several tools/package enable such inferences i.e. EstimateS…)

Line 523 : how far is vineyard 2 from the cellar (or from any other) in comparison to vineyard 1?

Line 567-72 Table 5 : same comment as for table 4.

Line 615-31: same as above : which dissimilarity distance has been used for PCoA

Line 654 : “genetically similar to the commercial strain Velluto BMV58” . Could the authors estimate a likelihood to obtain the same genotype by chance from the frequencies of each allele in this dataset ?

Line 663-667 : more than the number of heterozygote loci per strains , or heterozygote strains, I think that it is more relevant to infer a selfing rate. It takes into account the allelic richness of the loci and make the comparison easier for different datasets.

Line 703-707 : Some admixing events is far from a panmictic population. And indeed as for S. cerevisiae, S. uvarum has been described as an inbred species. The extent of admixture should be evaluated. See comment above about the interest of methods inferring admixed populations. The addition of a figure presenting this population structure may also explain uneven clustering of strains in the tree according to clusters.

The tree does not contain bootstrap value, which may also explain this pattern . The robustness of these nodes could be evaluated by a jacknife or a bootstrap procedure.

Reviewer #2: Review for POND-D-19-30860

This is an interesting study looking at microbial diversity in Canadian wine systems. The main issue with the study as it stands is that appropriate analyse have not been conducted on the data and thus some conclusions are not supported. Specifically, the authors make the following statements in the conclusion section that are not supported by analyses:

Differences in fungal community composition were observed, with H. osmophila representing a significant proportion of the fungal community in the fermentations from one vineyard, but not the other.

>no significance is shown

The presence of persistent non-commercial strains, as well as the phylogenetic tree generated to compare Okanagan and international strains, provides evidence for an indigenous winery-resident S. uvarum population in the Okanagan Valley of Canada

>no evidence for indigenous winery-resident S. uvarum population is provided.

The study deserves publication after these, and other less major issues listed below, are dealt with.

Other comments

Line 53-54 “aggressively competitive towards indigenous yeasts” – a couple of points here: 1) I think it is the massive inoculation size of commercial yeast that allows them to outcompete indigenous yeasts – not some other inherent advantage; 2) ‘competitive towards’ is not grammatically correct ‘more competitive than..’ would work.

The authors cite RStudio for analyses – I appreciate they used this (as do I) but the R platform was used for analyses – Rstudio is just an interface to R.

How many tanks were each vineyard put into? Just one? If so it isn’t really appropriate to use the sub-samples from this mixed tank as replicates – these are pseudo samples – one would need to sample multiple tanks. Either way, given the low sample numbers and non-normality of the chemical data, isn’t it more conservative and thus appropriate to use a Kruskal-Wallis text rather than an ANOVA?

I think we need clarification of whether any YAN or sulphur additions were made to the juices and if so what these were.

The logic to not perform inferential statistics on the fungal community data – due to different times of harvest and fermentation as well as location is not sound. The statistical tests are the only way to test if these communities differ – if they do one cannot determine whether location or time (or both) drove this difference – but this does not mean this should not be tested… Both between tank and within tank statistical analyses of fungal communities are absolutely required to support any statements of difference and change. Phrases such as ‘very different’ are not valid unless the are supported by an analyses. This is the core of science – not subjective interpretation but by means of objective analyses. Such analysis needs to be applied to the whole of the ‘Fugal communities’ section on the results. If not, this entire section is simply an observation and no conclusions may be drawn or made.

Some basic presentation of the fungal community data is desirable: how many reads and ASVs (species) were there? What classes, orders etc did these encapsulate?

“S. uvarum, which dominated the fermentations from both vineyards, is a glucophilic

yeast [29], and the residual fructose observed in the late stage of fermentation suggests that this yeast is less adept at fermenting fructose than S. cerevisiae. This result is supported by previous research [55]” and other discussion about yeasts in lines 410-413 - the authors have not yet shown what species dominated or are present the ferments they analysed – it would be better to move these statements until after they have.

‘two of the dominant strains were previously identified as dominant strains

39 in uninoculated Chardonnay fermentations at the same winery two years earlier,

40 providing evidence for a winery-resident population of indigenous S. uvarum’

and

Line 590 “The reoccurrence of ‘2015 Strain 2’ and ‘2015 Strain 3’ suggests that these strains have established themselves as persistent winery residents, capable of entering and dominating fermentations in multiple vintages.”

I don’t see why this conclusion is drawn and what support it? Why is this conclusion draw over the other option – that these strains reside in the vineyards the fruit derived from – that is the reason they reoccur… there is good evidence that a large genetic variability of Saccharomyces yeasts reside in various vineyard habitats (soil, bark) and that these are regionally distinct populations – what direct evidence is there that live Saccharomyces overwinter in wineries?

Since yeast are sexual (i.e. loci recombine – the authors show heterozygosity in their data) is there a reason the authors did not analyse the microsatellite data with a method that accounts for this – some network analyses for example? The comparison of individual stains is not that powerful or appropriate – it is the degree to which the populations are genetically related and structured by analysing the alleles they share that is the better approach. Why was a ‘Structure’ analyses or similar not conducted?

Why is a Bruvo distance of 0.3 ‘genetically similar’ – what does this mean? Can the authors quantify this in some way?

6. PLOS authors have the option to publish the peer review history of their article (what does this mean?). If published, this will include your full peer review and any attached files.

Reviewer #1: Yes: JL Legras

Reviewer #2: No

---

## [Author Response · Author response to Decision Letter 0]

15 Jun 2020

Reviewer and Editor comments:

*Please note that references to line numbers in the manuscript refer to the cleaned and highlighted version, not the track-changes version.

Associate Editor

Thank you for submitting your manuscript to PLOS ONE. After careful consideration, we feel that it has merit but does not fully meet PLOS ONE’s publication criteria as it currently stands. Therefore, we invite you to submit a revised version of the manuscript that addresses the points raised during the review process, in particular the following major points raised by referees: 1- S. cerevisiae x S. uvarum hybrid detection, 2- the use of another software than adegenet for the detection of admixture, 3- statistical issues raised by second referee, such as Kruskal-Wallis text vs ANOVA, and statistical evaluation of communities, 4- taking into account heterozygosity at microsatellite loci. All minor points raised by referees should be addressed as well. 

1. S. cerevisiae x S. uvarum hybrids:

a. A discussion of the potential for S. cerevisiae x S. uvarum hybrids in this population has been added on Lines 892-916. In this manuscript, we only present strains for which all 11 microsatellite loci amplify with S. uvarum primers, whereas typically hybrid strains have a reduced success rate of microsatellite amplification with a primer set for one particular species. In a previous study conducted in 2015 at the same winery (Morgan et al. 2019, DOI: 10.1093/femsyr/foz049), yeast isolates were initially presumed to belong to S. cerevisiae, but no amplification was detected with S. cerevisiae microsatellite primers, which is what initially led us to test S. uvarum primers on these isolates. Therefore, we know conclusively that the 66 strains identified in both the current study and the 2015 study are not hybrids with S. cerevisiae. Future studies targeting the presence of hybrid strains will help us determine if they do indeed constitute an important component of the yeast community at this winery, but investigating hybrid strains was not an objective in this study.

b. The Wallerstein agar was determined previously to show a definitive colour difference between these two species (Morgan et al. 2020, DOI: 10.5344/ajev.2020.19072; also see Figure S2, newly added to this manuscript). Therefore, we chose this method to differentiate between presumed S. cerevisiae and presumed S. uvarum isolates prior to identification via microsatellite analysis.

c. We are currently conducting whole genome sequencing of five of the strains identified in this study. We will be mapping the sequencing reads against the S. uvarum reference strain CBS 7001 in the coming month, which will allow us to conclusively determine how much of the genomes of these strains maps to S. uvarum, and what introgressions there may be from other Saccharomyces species. However, the whole genome sequencing project is beyond the scope of this current study.

2. At the request of both reviewers, we have added an analysis of population structure of the strains of significance as compared to international strains, using InStruct software and visualized with DISTRUCT plots. Related discussions can be viewed on Lines 808-848, and the plot is presented in a new figure (Fig 5).

3. Statistical analysis of the fungal communities and S. uvarum populations have been added. Details of the statistical tests are described in the completely-revised Statistical Analysis section of the Materials and Methods (Lines 337-429), while a discussion of statistical results has been added throughout the Results and Discussion section (Lines 496-499; 525-527; 595-598; 663-664; 729-732; and Tables 3, 4, and 5). We have chosen not to perform non-parametric tests such as Kruskal-Wallis tests, and instead to keep the parametric ANOVA tests, because our data met the assumption of homogeneity of variance, and ANOVA are robust to deviations from normality. We have explained our reasoning on Lines 342-344.

4. We have expanded our discussion of the genetic diversity and population structure of this yeast population, including fixation indices and observed vs. expected heterozygosity, on Lines 780-791; 803-807; 808-848.

Reviewer #1

In this manuscript McCarthy et al. present the characterization of a unique yeast population from a Canadian winery. Interestingly, this winery is inhabited by a S. uvarum population, which makes it one of the few example of yeast microflora not containing Saccharomyces cerevisiae. It presents a comprehensive analysis of the fungal population from the vineyard to the vats, and an analysis of S. uvarum population in the winery and the vineyard.

This original, well led and well written study is an interesting case study of alternative micro-flora under cold climate. This case was known for cool climate vineyards, but until now, this is the first study presenting this situation.

Despite the quality of the manuscript, I feel one main lack and few missing piece of information or analyses.

1 – The authors have selected theirs strains on Wallerstein nutrient media. This is not a robust species identification method, and authors have not explored for the presence of S. cerevisiae x S. uvarum interspecific hybrids that may participate to the ecosystem. Doing so they may have omitted one component of the fermentation microflore. This had been described in one of the first winery containing S. uvarum strains (Demuyter et al. 2014 J. Applied Microbiol , LeJeune et al. 2007 FEMS Yeast Research), and I personally have detected many others in cold cellars. As a consequence, it is necessary to check for the presence of hybrids among the “S. uvarum” isolates (that microsatellite considered as S. uvarum) or among the S. cerevisiae. An overview of the frequencies of the counts of S. uvarum and S. cerevisiae in the different samples is missing.

• The Wallerstein agar was determined previously to show a definitive colour difference between these two species (Morgan et al. 2020, DOI: 10.5344/ajev.2020.19072; also see Figure S2, newly added to this manuscript). Therefore, we chose this method to differentiate between presumed S. cerevisiae and presumed S. uvarum isolates prior to identification via microsatellite analysis.

• We have added a discussion of the potential for S. cerevisiae x S. uvarum hybrids in this population on Lines 892-916. In this manuscript, we only present strains for which all 11 microsatellite loci amplify with S. uvarum primers, whereas typically hybrid strains have a reduced success rate of microsatellite amplification with a primer set for one particular species. In a previous study conducted in 2015 at the same winery (Morgan et al. 2019, DOI: 10.1093/femsyr/foz049), yeast isolates were initially presumed to belong to S. cerevisiae, but no amplification was detected with S. cerevisiae microsatellite primers, which is what initially led us to test S. uvarum primers on these isolates. Therefore, we know that the 66 strains identified in both the current study and the 2015 study are not hybrids with S. cerevisiae. We did not test the isolates that only partially amplified using S. uvarum primers with S. cerevisiae primers; future studies targeting the presence of hybrid strains will help us determine if they do indeed constitute an important component of the yeast community at this winery. However, the targeting of hybrids was not an objective of this study.

• We have added a discussion of the LeJeune et al. 2007 paper that looked at hybrids (Lines 905-907), and we believe the Demuyter et al. 2014 paper mentioned was a typo, and refers to the Demuyter et al. 2004 paper, which is referenced throughout the manuscript.

• To acknowledge that there is the potential for hybrid or mosaic genomes, we have used the phrase ‘presumed S. cerevisiae’ and ‘presumed S. uvarum’ when referring to pre-strain typed yeast isolates. We have included two new figures (Fig 2 and Fig S2) that show the relative proportion of presumed S. cerevisiae and presumed S. uvarum strains in the fermentations, as estimated by visual differentiation on WLN media. The presumed S. cerevisiae isolates were in fact strain-typed (strain identities available on OSF: https://osf.io/j7rx8/). A discussion of these results has been added to Lines 648-652.

2 – When analyzing their population structure, with the package adegenet, the authors mention that the topology of the tree was not matching the clustering obtained with the “find.cluster function of the adegenet package. This function is very efficient for the identification of groups, but less for the identification of admixed populations. Alternative populations structure software enable such inferences : i.e. InStruct, or Structure softwares take into account admixture, making the assumption of an inbred population (described as the case in manuscripts of other groups) or maximizing the likelihood of Hardy Weinberg equilibrium.

• We have added an analysis of population structure of the strains of significance as compared to international strains, using InStruct software and visualized with DISTRUCT plots. Related discussions can be viewed on Lines 808-848, and the plot can be seen in Fig 5.

• We have also removed any statements regarding admixed or panmictic populations in reference to the tree (Fig 6), which has been updated with bootstrap values.

3 – Some minors point

Line 127-147 : what is the total weight of the grape samples?

• The total weight of the grape samples was 2-3 kg (Line 142).

Line 364: the distance used for the estimation of dissimilarity is not indicated: the function “vegdist” of the vegan package uses several dissimilarity distances (i.e : Bray-Curtis, Jaccard…)

• We apologize for this unintended omission. We have re-written the entire Statistical Analysis section, and have included this information. We used a Bray-Curtis dissimilarity matrix when performing PERMANOVA/PERMDISP tests (Lines 372, 376) and generating the PCoA ordination (Line 389).

Line 424 : the unit in which volatile acidity is expressed is not indicated : is this g/L of acetic acid, or H2SO4? This threshold is very high and caused by the choice of the analytical method. Other methods (i.e. enzymatic kits, or distillation) provide much higher sensitivity.

• Estimated as g/L acetic acid (added to Line 485).

• In this context, ‘detection threshold’ refers to the sensory (tasting) threshold, not an analytical detection threshold. This has been clarified in text (Line 488).

Line 439-449: Table 4 : Why only Simpson’s Index is given. Other indices such as the popular Shannon index provide complementary information on the diversity isolated in the samples. A rarefaction method should be used in order to avoid variations of sample size (several tools/package enable such inferences i.e. EstimateS…)

• We have added Shannon’s Index to Table 4 and Table 5. We initially chose Simpson’s index because it places more emphasis on dominant species/strains, which is what we were most interested in.

• We have created rarefaction curves for both the fungal species and S. uvarum strains (Lines 348-356; Fig S3). However, our data did not include variations of sample size during analysis, because all samples were rarefied prior to analysis.

Line 523 : how far is vineyard 2 from the cellar (or from any other) in comparison to vineyard 1?

• The two vineyards are directly south of the winery and separated from each other by approximately 30 km. This information has been added to Lines 119-121.

Line 567-72 Table 5 : same comment as for table 4.

• We have added Shannon’s Index to Table 4 and Table 5.

Line 615-31: same as above : which dissimilarity distance has been used for PCoA

• We used a Bray-Curtis dissimilarity matrix when performing PERMANOVA/PERMDISP tests (Lines 372, 376) and generating the PCoA ordination (Line 389).

Line 654 : “genetically similar to the commercial strain Velluto BMV58” . Could the authors estimate a likelihood to obtain the same genotype by chance from the frequencies of each allele in this dataset ?

• We have used GenAlEx to calculate probability of identity (PI) - the probability that two unrelated individuals would have identical multilocus genotypes. PI was calculated to be 6.7 � 10-8. However, this probability does not take into account the Bruvo distance used in this study; individuals in this study could have slightly different MLGs and still be grouped into the same strain classification. This discussion has been added to Lines 780-784.

Line 663-667 : more than the number of heterozygote loci per strains , or heterozygote strains, I think that it is more relevant to infer a selfing rate. It takes into account the allelic richness of the loci and make the comparison easier for different datasets.

• We have used GenAlEx to calculate fixation indices (FST and FIS) which estimate the variance in allele frequency between subpopulations and inbreeding coefficients, respectively. FST was calculated to be 0.014 ± 0.003, indicating that the two subpopulations (Vineyard 2 and Vineyard 8) share a high degree of genetic material, and suggestings a high level of breeding between the subpopulations. FIS was calculated to be 0.74 ± 0.04, indicating a considerable degree of inbreeding. This information has been added to Lines 784-791.

• We have also used GenAlEx to compare the observed heterozygosity rate, and found it to be 2 to 16 times lower than the unbiased expected heterozygosity rate for each locus, suggesting that this population has a high selfing rate (added to Lines 803-807).

Line 703-707 : Some admixing events is far from a panmictic population. And indeed as for S. cerevisiae, S. uvarum has been described as an inbred species. The extent of admixture should be evaluated. See comment above about the interest of methods inferring admixed populations. The addition of a figure presenting this population structure may also explain uneven clustering of strains in the tree according to clusters.

The tree does not contain bootstrap value, which may also explain this pattern. The robustness of these nodes could be evaluated by a jacknife or a bootstrap procedure.

• We have added an analysis of population structure of the strains of significance as compared to international strains, using InStruct software and visualized with DISTRUCT plots. Related discussions can be viewed on Lines 808-848, and the plot can be seen in Fig 5.

• We have also removed any statements regarding admixed or panmictic populations in reference to the tree, and have added bootstrap values above 0.5 to Fig 6.

Reviewer #2

This is an interesting study looking at microbial diversity in Canadian wine systems. The main issue with the study as it stands is that appropriate analyse have not been conducted on the data and thus some conclusions are not supported. Specifically, the authors make the following statements in the conclusion section that are not supported by analyses:

Differences in fungal community composition were observed, with H. osmophila representing a significant proportion of the fungal community in the fermentations from one vineyard, but not the other.

>no significance is shown

• We have completely re-written the Statistical Analysis section, and have added statistical analyses of the fungal communities and S. uvarum populations in this study, described on Lines 337-429. A discussion of statistical results has been added throughout the Results and Discussion section (Lines 496-499; 525-527; 595-598; 663-664; 729-732; and Tables 3, 4, and 5).

• Data analyses conducted in the revised manuscript include: (1) statistical comparisons of chemical parameters, alpha diversity (Simpson’s and Shannon’s Indices), and beta diversity (composition, based on Bray-Curtis dissimilarity); (2) rarefaction curves of species/strain richness; (3) S. uvarum population structure using InStruct; (4) unrooted tree of S. uvarum strains using k-means clustering with added bootstrap values; (5) estimations of selfing, (in)breeding, and probability of identity within the S. uvarum population.

• We have removed the term ‘significant’ when discussing the proportion of H. osmophila in the conclusion. However, we do not believe statistics are necessary in order to discuss the differential abundance of this species in the two vineyard treatments, which can be visually distinguished in Fig 1, and which is a small point of discussion.

The presence of persistent non-commercial strains, as well as the phylogenetic tree generated to compare Okanagan and international strains, provides evidence for an indigenous winery-resident S. uvarum population in the Okanagan Valley of Canada

>no evidence for indigenous winery-resident S. uvarum population is provided.

• We agree that we do not have conclusive proof for an indigenous winery-resident population. However, we did detect an overlap of 66 strains in Chardonnay fermentations at the same winery two years apart. This overlap is much higher than has been detected from studies analyzing vineyard yeast populations from successive vintages. Therefore, we suspect that the majority of strains identified in this study reside in the winery, but the non-overlapping strains (especially those with differential abundance in the fermentations from one vineyard over another) could be derived from the vineyard. We have softened the language of the Abstract and Conclusion sections (Lines 40; 956-957) and added a discussion of winery- versus vineyard-resident yeasts, taking into account both our data and previous research on this topic (Lines 618-628; 687-690; 717-720).

The study deserves publication after these, and other less major issues listed below, are dealt with.

Other comments

Line 53-54 “aggressively competitive towards indigenous yeasts” – a couple of points here: 1) I think it is the massive inoculation size of commercial yeast that allows them to outcompete indigenous yeasts – not some other inherent advantage; 2) ‘competitive towards’ is not grammatically correct ‘more competitive than..’ would work.

• We have changed the wording of this statement (Line 53). However, previous research supports the idea that commercial yeast strains used previously at a winery may out-compete indigenous strains, even when they are not inoculated. This discussion has been added to Lines 54-56.

The authors cite RStudio for analyses – I appreciate they used this (as do I) but the R platform was used for analyses – Rstudio is just an interface to R.

• We have amended this wording (Lines 225; 338).

How many tanks were each vineyard put into? Just one? If so it isn’t really appropriate to use the sub-samples from this mixed tank as replicates – these are pseudo samples – one would need to sample multiple tanks. Either way, given the low sample numbers and non-normality of the chemical data, isn’t it more conservative and thus appropriate to use a Kruskal-Wallis text rather than an ANOVA?

• Must from each vineyard was first transferred temporarily to a large stainless steel settling tank, before being transferred to stainless steel barrels. Three stainless steel barrels per vineyard were analyzed in this study, and the cold settling samples for microbial and chemical analysis were all taken from the stainless steel barrels. We believe our use of ‘cold settling tank’ has resulted in confusion, and have changed the name of the tank in the manuscript text to simply a ‘settling tank’ (Lines 156; 158; 159). In this study, ‘cold settling’ refers to the period of time before alcoholic fermentation started.

• We have chosen not to perform non-parametric tests such as Kruskal-Wallis tests, and instead to keep the parametric ANOVA tests, because our data met the assumption of homogeneity of variance, and ANOVA are considered robust to deviations from normality. We have explained and cited our reasoning on Lines 342-344.

I think we need clarification of whether any YAN or sulphur additions were made to the juices and if so what these were.

• This information is present on Lines 169-173.

The logic to not perform inferential statistics on the fungal community data – due to different times of harvest and fermentation as well as location is not sound. The statistical tests are the only way to test if these communities differ – if they do one cannot determine whether location or time (or both) drove this difference – but this does not mean this should not be tested… Both between tank and within tank statistical analyses of fungal communities are absolutely required to support any statements of difference and change. Phrases such as ‘very different’ are not valid unless the are supported by an analyses. This is the core of science – not subjective interpretation but by means of objective analyses. Such analysis needs to be applied to the whole of the ‘Fugal communities’ section on the results. If not, this entire section is simply an observation and no conclusions may be drawn or made.

• We have completely re-written the Statistical Analysis section, and have added statistical analyses of the fungal communities and S. uvarum populations in this study, described on Lines 337-429. A discussion of statistical results has been added throughout the Results and Discussion section (Lines 496-499; 525-527; 595-598; 663-664; 729-732; and Tables 3, 4, and 5).

• Data analyses conducted in the revised manuscript include: (1) statistical comparisons of chemical parameters, alpha diversity (Simpson’s and Shannon’s Indices), and beta diversity (composition, based on Bray-Curtis dissimilarity); (2) rarefaction curves of species/strain richness; (3) S. uvarum population structure using InStruct; (4) unrooted tree of S. uvarum strains using k-means clustering with added bootstrap values; (5) estimations of selfing, (in)breeding, and probability of identity within the S. uvarum population.

Some basic presentation of the fungal community data is desirable: how many reads and ASVs (species) were there? What classes, orders etc. did these encapsulate?

• We have added a basic presentation of the fungal community data on Lines 517-524. Our fungal community data were rarefied to 20,000 sequences per sample and presented as relative abundance.

“S. uvarum, which dominated the fermentations from both vineyards, is a glucophilic

yeast [29], and the residual fructose observed in the late stage of fermentation suggests that this yeast is less adept at fermenting fructose than S. cerevisiae. This result is supported by previous research [55]” and other discussion about yeasts in lines 410-413 - the authors have not yet shown what species dominated or are present the ferments they analysed – it would be better to move these statements until after they have.

• We have moved these statements to another section (moved to Lines 636-639).

‘two of the dominant strains were previously identified as dominant strains

39 in uninoculated Chardonnay fermentations at the same winery two years earlier,

40 providing evidence for a winery-resident population of indigenous S. uvarum’

and

Line 590 “The reoccurrence of ‘2015 Strain 2’ and ‘2015 Strain 3’ suggests that these strains have established themselves as persistent winery residents, capable of entering and dominating fermentations in multiple vintages.”

I don’t see why this conclusion is drawn and what support it? Why is this conclusion draw over the other option – that these strains reside in the vineyards the fruit derived from – that is the reason they reoccur… there is good evidence that a large genetic variability of Saccharomyces yeasts reside in various vineyard habitats (soil, bark) and that these are regionally distinct populations – what direct evidence is there that live Saccharomyces overwinter in wineries?

• As described above, we based our conclusions on previous research showing very little overlap between vineyard S. cerevisiae populations from vintage to vintage (Martiniuk et al. 2016; DOI: 10.1371/journal.pone.0160259). We have softened the language of the Abstract and Conclusion sections (Lines 40; 956-957) and added a discussion of winery- versus vineyard-resident yeasts, taking into account both our data and previous research on this topic (Lines 618-628; 687-690; 717-720).

• There is a body of evidence that Saccharomyces and other yeasts overwinter in wineries (discussion added on Lines 625-628), and the presence of certain dominant yeast strains in fermentations from vineyards that are separated by more than 30 km (and did not enter the winery at the same time) suggests that many of these yeast strains are winery-residents. However, it is possible that some of these yeasts may live in the vineyard (and it is of course likely that this population originated in a nearby natural environment), and we have added statements to this effect (Lines 618-620; 717-720.

Since yeast are sexual (i.e. loci recombine – the authors show heterozygosity in their data) is there a reason the authors did not analyse the microsatellite data with a method that accounts for this – some network analyses for example? The comparison of individual stains is not that powerful or appropriate – it is the degree to which the populations are genetically related and structured by analysing the alleles they share that is the better approach. Why was a ‘Structure’ analyses or similar not conducted?

• We have added an analysis of population structure of the strains of significance as compared to international strains, using InStruct software and visualized with DISTRUCT plots. Related discussions can be viewed on Lines ##-##, and the plot can be seen in Fig 5.

Why is a Bruvo distance of 0.3 ‘genetically similar’ – what does this mean? Can the authors quantify this in some way?

• Bruvo distance is calculated using an algorithm that takes into account stepwise mutations, making it appropriate for use with microsatellite data. This distance is calculated on experimental data and allows the user to collapse MLGs with slight differences in allele size into a single strain category, based on similarity at a threshold value from 0 to 1. An applied threshold of 0 results in every unique MLG being classified as a different strain, and an applied threshold of 1 results in all the MLGs in a dataset being collapsed into a single strain. This information has been added to the manuscript on Lines 218-224.

---

## [Decision Letter · Decision Letter 1]

20 Jul 2020

PONE-D-19-30860R1

An indigenous Saccharomyces uvarum population with high genetic diversity dominates uninoculated Chardonnay fermentations at a Canadian winery

PLOS ONE

Dear Dr. Morgan,

Thank you for submitting your manuscript to PLOS ONE. After careful consideration, we feel that it has merit but does not fully meet PLOS ONE’s publication criteria as it currently stands. Therefore, we invite you to submit a revised version of the manuscript that addresses the points raised during the review process.

Both reviewers felt that the ms has been greatly improved, but they have made several additional comments that all need to be addressed. Reviewer 1 mentions additional experiments. You may choose not to do these experiments, but in that case, you should modify your text to take into account the possibility mentioned by the reviewer that you may have missed the observation of hybrids.

We look forward to receiving your revised manuscript.

Kind regards,

Cecile Fairhead, Ph.D.

Academic Editor

PLOS ONE

Reviewers' comments:

Reviewer's Responses to Questions

**Comments to the Author**

1. If the authors have adequately addressed your comments raised in a previous round of review and you feel that this manuscript is now acceptable for publication, you may indicate that here to bypass the “Comments to the Author” section, enter your conflict of interest statement in the “Confidential to Editor” section, and submit your "Accept" recommendation.

Reviewer #1: (No Response)

Reviewer #2: All comments have been addressed

2. Is the manuscript technically sound, and do the data support the conclusions?

Reviewer #1: Partly

Reviewer #2: Yes

3. Has the statistical analysis been performed appropriately and rigorously? 

Reviewer #1: No

Reviewer #2: Yes

4. Have the authors made all data underlying the findings in their manuscript fully available?

Reviewer #1: No

Reviewer #2: Yes

5. Is the manuscript presented in an intelligible fashion and written in standard English?

Reviewer #1: Yes

Reviewer #2: Yes

6. Review Comments to the Author

Reviewer #1: This new version of the manuscript of Morgan et al. is clearly improved in many aspects, and I appreciated the new statistical analyses that were provided. Nevertheless, I have still several comments to make.

1. About the potential hybrid status of the strains, I disagree with the authors on several points:

“a. In this manuscript, we only present strains for which all 11 microsatellite loci amplify with S. uvarum primers, whereas typically hybrid strains have a reduced success rate of microsatellite amplification with a primer set for one particular species.”

This statement is not right. Some hybrids have indeed uneven contribution of each parental genome in their genome, but often the contribution of each parent remain significant and similar (i.e. Le Jeune al. 2007 for microsatellite typing of both species)

The authors have characterized 66 strains in 2015 which share their genotypes with some actual isolates. The same genotypes from the S. uvarum genome side does not mean that all isolates are S. uvarum only. Indeed, in Lejeune et al. 2007, a hybrid strain and a potential parental strain were found with the same genotype. In addition, we can see from the tree in figure 6 (that contain 31 of these strains) that these genotypes are not spread evenly in the S. uvarum tree: some 2015 genotypes belong to specific clusters of strains, so that some clusters might not be explored.

“b. The Wallerstein agar was determined previously to show a definitive color difference between these two species (Morgan et al. 2020, DOI:10.5344/ajev.2020.19072; also see Figure S2, newly added to this manuscript)”

This is an interesting point, but we do not know what would the color of the colony of a hybrid on this media: closer to one of the other parent or between the two colors.…

“c. We are currently conducting whole genome sequencing of five of the strains identified in this study. We will be mapping the sequencing reads against the S. uvarum reference strain CBS 7001 in the coming month…” This an interesting point, but one can wonder what is the position of these sequenced strains among the groups of strains that have been genotyped.

As a summary, and even if the presence of hybrids is not the focus of that paper, these 3 answers are not really solid enough in order to put aside the presence of hybrids in this winery. Because they may combine the cold temperature fermentation ability and the performance of S. cerevisiae, they have the potential for a better fitness in such an environment, especially in a cellar. I am reluctant to ask for more experimental work, but ITS amplification (from cell suspension obtained from colonies) + digestion with HaeIII is a very simple control to perform that would provide a relevant and solid information about the presence of hybrids, and their potential ability to colonize the winery.

Some minor comments :

Page 19 line 120 : “Vineyard 8 is located approximately 90 km south of the winery; the two wineries are” : this should be the two vineyards…

Page 23 line 220-226 : The explanation of the meaning of a threshold is clear, but the criteria leading to the choice of 0.3 are not : why 0.3 is better than 0.1 or .01. This has may be been inferred from a bimodal distribution of the values of the pairwise distance matrix …. Please explain your choice (see reviewer 2)

Page 30, line 371: a reference should be given for the PERMANOVA method

Page 31, line 712-720 : The authors cannot rule-out the presence of specific populations in the fermentation from different vineyards as the result of the import of two different S. uvarum population.

Page 32, line 422-3 : this k-means clustering is a tool developed with the DAPC method (Discriminant analysis of principal components), described in Jombart et al. 2010 BMC Bioinformatics, 11:94, that should be cited.

Page 33, line 769 -779 : It is really difficult to conclude to a relatedness from an arbitrary threshold of 0.3 As mentioned earlier if we had an idea of the distribution of the pairwise distances obtained with this metric, that would enable to figure out the distribution of distances which might give more weight. The authors should have in mind the amazing high relatedness found among strains from the Holartic group for this species (See Almeida et al. 2014 Nature Comm). I think it might be useful to highlight in the tree may be the presence of natural isolates from Canada or USA.

Page 50, line 850- 874: This paragraph is interesting but raises questions. I had not fully seen the problem of sub-setting the data for this analysis in the first version of the manuscript.

1- I think that the number of cluster inferred by InStruct is amazingly high: especially as most strains appear as highly admixed. There might have been a problem to reach convergence of the MCMC chain or InStruct had difficulties to reach a convergence because of the data. This could come from the small number of individuals given the information contained in the dataset: too small to infer ancestry.

2- In addition, for some data sets, InStruct infers higher values of K but at the cost of much more variable of DIC. I would recommend plotting DIC according to K. A first plateau with little variance of DIC might appear before the lowest value of DIC. A higher variance among chains should be visible from the 5 chains

The authors should check for a possible solution with a lower number of ancestral population. The barplot suggests indeed the presence of 6 major groups (K1, K2, K3, K10, K4, K8). At least the solution K=5 should be drawn in order to permit comparison with the clustering obtained with DAPC.

3- In the same manner, the clustering with DAPC, has apparently been performed on a subset of strains including 12 from other origins.

There is no reason to perform a tree, an ancestry analysis and a clustering with a subset of “relevant strains". What may look technologically relevant may not be ecologically relevant. In addition, given the high number of strains obtained, the frequency of many genotypes may be underestimated while others are overestimated so why choose 1%. A tree containing 106 + 12 extra strains should be easy to draw and color with the "ggtree" R package and the "groupOTU" function. This tree might also show as well how related are the different strains.

Nevertheless, the identification of the specificities of the Okanagan population is indeed amazing, and should appear.

Data: the microsatellite dataset should be made available

Reviewer #2: Review of PONE-D-19-30860R1

I commend the authors for a serious revision of the manuscript in light if review comments – it is nice to see serious revisions rather than minimal changes.

I think this represents a solid study, and have only very minor further suggestions.

Can the R^2 values be included when PermANOVA results are presented?

Can 95% confidence limits be included with the estimation of proportions of different species (lines 599-604)

Lines 608 – 610 “identification techniques such as high-throughput amplicon sequencing have allowed for the identification of all microorganisms present in wine fermentations…”

Even very deep sequencing will unlikely get ‘all’ species; this should be changed to ‘most’ or some equivalent.

Lines 600 - 615

The authors need to appreciate that this method analyses DNA – not live cells – from samples: the presence of H. osmophila DNA does not mean these cells are alive or even intact…

On that note – did the ITS1 ASVs you are using to count H. osmophila allow you to identity to species level with such certainly? Usually ITS1 can get to genus robustly, but not always species level? The same comment for other species identified with ITS1 sequences.

7. PLOS authors have the option to publish the peer review history of their article (what does this mean?). If published, this will include your full peer review and any attached files.

Reviewer #1: No

Reviewer #2: **Yes: **Matthew R Goddard

---

## [Author Response · Author response to Decision Letter 1]

31 Oct 2020

Response to Reviewers

*Please note that references to line numbers in the manuscript refer to the cleaned version, not the track-changes version.

Reviewer #1: 

This new version of the manuscript of Morgan et al. is clearly improved in many aspects, and I appreciated the new statistical analyses that were provided. Nevertheless, I have still several comments to make.

1. About the potential hybrid status of the strains, I disagree with the authors on several points:

“a. In this manuscript, we only present strains for which all 11 microsatellite loci amplify with S. uvarum primers, whereas typically hybrid strains have a reduced success rate of microsatellite amplification with a primer set for one particular species.”

This statement is not right. Some hybrids have indeed uneven contribution of each parental genome in their genome, but often the contribution of each parent remain significant and similar (i.e. Le Jeune al. 2007 for microsatellite typing of both species)

• We discussed the Le Jeune et al. 2007 paper on lines 702-705 and have clarified our viewpoint. We acknowledge that some hybrids have an equal contribution of genomes from each parent species, as is the case of the strains in the Le Jeune 2007 paper. However, based on our analysis of the ITS1 region of the rRNA gene by RFLP using HaeIII (described below in more detail under 1c), we have confirmed that the strains we have included in the analysis of our study are S. uvarum and not hybrids of S. uvarum and S. cerevisiae.

The authors have characterized 66 strains in 2015 which share their genotypes with some actual isolates. The same genotypes from the S. uvarum genome side does not mean that all isolates are S. uvarum only. Indeed, in Lejeune et al. 2007, a hybrid strain and a potential parental strain were found with the same genotype. In addition, we can see from the tree in figure 6 (that contain 31 of these strains) that these genotypes are not spread evenly in the S. uvarum tree: some 2015 genotypes belong to specific clusters of strains, so that some clusters might not be explored.

• At the request of the reviewer, we have amplified the ITS1 region of the rRNA gene on 50 of the most commonly identified strains and carried out a restriction enzyme digest with HaeIII. The results of this restriction digest are presented on lines 706-718 and Figure S4. (More detailed discussion of these results are presented below under 1c).

“b. The Wallerstein agar was determined previously to show a definitive color difference between these two species (Morgan et al. 2020, DOI:10.5344/ajev.2020.19072; also see Figure S2, newly added to this manuscript)”

This is an interesting point, but we do not know what would the color of the colony of a hybrid on this media: closer to one of the other parent or between the two colors.…

• We plated a known pure S. uvarum strain (CBS 7001), a known pure S. cerevisiae strain (Lalvin BA11), and a known S. uvarum x S. cerevisiae hybrid strain (Lalvin S6U, formerly referred to as a S. bayanus x S. cerevisiae hybrid) on WLN to determine the colour of the hybrid colony on this media. A picture of this plate has been added in Figure S2, and shows the S. uvarum strain as a green colony, the S. cerevisiae strain as a cream-coloured colony, and the hybrid strain as a cream-coloured colony. This suggests that WLN media is able to distinguish between pure S. uvarum strains and hybrid strains, but not between pure S. cerevisiae strains and hybrid strains (i.e. the hybrid colony appeared similar to the S. cerevisiae colony). The methods have been updated to reflect this addition on lines 206-211 and the discussion has been updated with this information on lines 686-693.

“c. We are currently conducting whole genome sequencing of five of the strains identified in this study. We will be mapping the sequencing reads against the S. uvarum reference strain CBS 7001 in the coming month…” This an interesting point, but one can wonder what is the position of these sequenced strains among the groups of strains that have been genotyped.

• The strains undergoing whole genome sequencing are as follows: 2015 Strain 1 (dominant in 2015), 2015 Strain 3 (dominant in 2015 and 2017), 2017 Strain 151 (dominant in 2017), 2015 Strain 163 (non-dominant), and 2017 Strain 097 (non-dominant). However, we have removed any references to whole genome sequencing because it is outside the scope of this current study.

As a summary, and even if the presence of hybrids is not the focus of that paper, these 3 answers are not really solid enough in order to put aside the presence of hybrids in this winery. Because they may combine the cold temperature fermentation ability and the performance of S. cerevisiae, they have the potential for a better fitness in such an environment, especially in a cellar. I am reluctant to ask for more experimental work, but ITS amplification (from cell suspension obtained from colonies) + digestion with HaeIII is a very simple control to perform that would provide a relevant and solid information about the presence of hybrids, and their potential ability to colonize the winery.

• At the request of the reviewer, we have conducted a digestion of the ITS1 region of the rRNA gene using the restriction enzyme HaeIII on 50 isolates from this study, representing the 50 most abundant strains. We have updated the Methods section to include this additional experiment (lines 259-269), and the results of this restriction digest are presented on lines 706-718 and Figure S4. These results show that the restriction patterns of the isolates match that of a pure S. uvarum strain, while the restriction patterns of the pure S. cerevisiae strain and the hybrid S. uvarum x S. cerevisiae strain used as control strains have distinct DNA fragment sizes. While we only had the resources to conduct this additional experiment on 50 of the 576 isolates analyzed in this study, these 50 isolates do represent the 50 most abundant strains. When combined with other evidence, such as the fact that in 2015 none of the isolates demonstrated amplification with S. cerevisiae microsatellite primers as well as the WLN agar colour selection, we are confident that the isolates analyzed in this study are pure S. uvarum strains, not hybrids.

• We acknowledge that there may be a low-level presence of hybrids in the winery environment, because we excluded a small number of isolates from our analysis when one or more loci consistently failed to amplify (described on lines 250-252 and discussed on lines 696-702). Further research specifically targeting hybrid populations will be able to determine if this winery, or the Okanagan valley in general, is host to a significant hybrid population, but this is outside the scope of our current study (lines 716-718).

Page 19 line 120 : “Vineyard 8 is located approximately 90 km south of the winery; the two wineries are” : this should be the two vineyards…

• This has been corrected (line 120).

Page 23 line 220-226 : The explanation of the meaning of a threshold is clear, but the criteria leading to the choice of 0.3 are not : why 0.3 is better than 0.1 or .01. This has may be been inferred from a bimodal distribution of the values of the pairwise distance matrix …. Please explain your choice (see reviewer 2)

• This explanation has been updated on lines 227-244. Based on poppr guidelines for selecting cut-off thresholds (https://grunwaldlab.github.io/poppr/articles/mlg.html#choosing-a-threshold), two methods may be used. The first, as the reviewer mentioned, is to select the threshold at the low point between a smaller peak and a larger peak, when the data have a bimodal distribution. In our case, the data are not bimodally distributed (see new Figure S3), so we chose the second method, which involves identifying ‘the largest gap between all putative thresholds.’ Based on Figure S3 we identified this cut-off to be approximately 0.3.

• Additionally, we chose this threshold to be consistent with the threshold used in a previous study of this same population (Morgan et al. 2019), because we were using strain identities determined in that previous study to strain-type isolates in this current study. Finally, because of the novelty of the high genetic diversity of the yeast population in this study (no other S. uvarum population has been found to be this genetically diverse to-date), it was important to us not to overestimate this diversity by using a lower threshold cut-off, which could artificially inflate the significance of our results. By using a conservative threshold, we are able to make more confident statements regarding the diversity of this population.

Page 30, line 371: a reference should be given for the PERMANOVA method

• We have included the citation for the ‘vegan’ package (which includes the PERMANVOA method ‘adonis’ that was used in this study) the first time it is mentioned, on line 376.

Page 31, line 712-720 : The authors cannot rule-out the presence of specific populations in the fermentation from different vineyards as the result of the import of two different S. uvarum population.

• We agree that we cannot rule out the presence of vineyard-derived yeast strains contributing to the observed populations, and have clarified our discussion of this topic on lines 789-793.

Page 32, line 422-3 : this k-means clustering is a tool developed with the DAPC method (Discriminant analysis of principal components), described in Jombart et al. 2010 BMC Bioinformatics, 11:94, that should be cited.

• We have removed k-means clustering from this manuscript and replaced with the clustering output from InStruct (Methods updated on lines 455-459).

Page 33, line 769 -779 : It is really difficult to conclude to a relatedness from an arbitrary threshold of 0.3. As mentioned earlier if we had an idea of the distribution of the pairwise distances obtained with this metric, that would enable to figure out the distribution of distances which might give more weight. The authors should have in mind the amazing high relatedness found among strains from the Holartic group for this species (See Almeida et al. 2014 Nature Comm). I 

• We have explained our reasoning for selecting a cutoff of 0.3 above and on lines 227-244 and Figure S3 of the manuscript. Additionally, we note that due to the extraordinarily high number of multilocus genotypes that were identified in this study, the cutoff threshold chosen could help reduce noise or potential errors in the genotyping method, which is a function of the poppr package.

Page 50, line 850- 874: This paragraph is interesting but raises questions. I had not fully seen the problem of sub-setting the data for this analysis in the first version of the manuscript.

1- I think that the number of cluster inferred by InStruct is amazingly high: especially as most strains appear as highly admixed. There might have been a problem to reach convergence of the MCMC chain or InStruct had difficulties to reach a convergence because of the data. This could come from the small number of individuals given the information contained in the dataset: too small to infer ancestry.

• We have re-run the InStruct structure analysis (lines 426-451) and re-created the DISTRUCT plot (Figure 5) to include all strains identified in this study, as well as the international strains. According to the DIC, K = 11 was found to be the optimal number of clusters. A very small minimum was also seen at K = 5, so we ran both levels of clustering through CLUMPP and visualized the output of both clustering thresholds using DISTRUCT (Figure 5A and 5B).

2- In addition, for some data sets, InStruct infers higher values of K but at the cost of much more variable of DIC. I would recommend plotting DIC according to K. A first plateau with little variance of DIC might appear before the lowest value of DIC. A higher variance among chains should be visible from the 5 chains

The authors should check for a possible solution with a lower number of ancestral population. The barplot suggests indeed the presence of 6 major groups (K1, K2, K3, K10, K4, K8). At least the solution K=5 should be drawn in order to permit comparison with the clustering obtained with DAPC.

• We included a solution for K = 5 clusters based on a DIC plot: the CLUMPP alignment was nearly optimal for K = 5 clusters ( H = 0.99) and was much lower for K = 11 (H = 0.60), so we accordingly focused most of our discussion on the K = 5 clustering method (lines 881-950).

3- In the same manner, the clustering with DAPC, has apparently been performed on a subset of strains including 12 from other origins.

There is no reason to perform a tree, an ancestry analysis and a clustering with a subset of “relevant strains". What may look technologically relevant may not be ecologically relevant. In addition, given the high number of strains obtained, the frequency of many genotypes may be underestimated while others are overestimated so why choose 1%. A tree containing 106 + 12 extra strains should be easy to draw and color with the "ggtree" R package and the "groupOTU" function. This tree might also show as well how related are the different strains.

Nevertheless, the identification of the specificities of the Okanagan population is indeed amazing, and should appear.

• We have re-created the phylogenetic tree (Figure 6) using all of the strains identified in this study, along with the 12 international strains, and clustered the tree according to the K = 5 output from the InStruct analysis described above. Strains with a dominant ancestor (defined as having a single inferred ancestor represent at least 75% of their total ancestry, determined by InStruct) were coloured in the tree accordingly. This has been updated in the Methods (lines 452-462) and Discussion (lines 963-991). We were able to identify Okanagan-specific clusters that were genetically determined by Bruvo distance to be different than the international strains.

Data: the microsatellite dataset should be made available

• The microsatellite genotypes of the strains identified at the winery in this study are in Table S3.

• The full microsatellite dataset is available at https://osf.io/j7rx8/. At this link, there is an Excel document titled ‘manuscript-data’ and within that document are two tabs containing microsatellite data: one tab contains only the relevant multilocus genotypes (MLGs) for this study (‘uvarum-mlgs-study’), and the other contains all the MLGs within our database (‘uvarum-database’).

Reviewer #2:

I commend the authors for a serious revision of the manuscript in light of review comments – it is nice to see serious revisions rather than minimal changes. I think this represents a solid study, and have only very minor further suggestions.

Can the R^2 values be included when PermANOVA results are presented?

• The R^2 values have been added to the PERMANOVA results (lines 556, 627, and 804).

Can 95% confidence limits be included with the estimation of proportions of different species (lines 599-604)

• All approximations/estimated proportions of species have been updated and are now represented as mean +/- SEM (to be consistent with values presented in Table S2) on lines 619-620 and 630-634.

Lines 608 – 610 “identification techniques such as high-throughput amplicon sequencing have allowed for the identification of all microorganisms present in wine fermentations…”

Even very deep sequencing will unlikely get ‘all’ species; this should be changed to ‘most’ or some equivalent.

• We have changed the wording of this statement to “most microorganisms” increase its accuracy (line 640).

Lines 600 - 615

The authors need to appreciate that this method analyses DNA – not live cells – from samples: the presence of H. osmophila DNA does not mean these cells are alive or even intact…

• We have updated this discussion to acknowledge the possibility that the DNA identified as belonging to H. osmophila may not have come from living cells (lines 646-649).

On that note – did the ITS1 ASVs you are using to count H. osmophila allow you to identity to species level with such certainly? Usually ITS1 can get to genus robustly, but not always species level? The same comment for other species identified with ITS1 sequences.

• ITS1 (fungi) has been generally very successful with getting to species-level identification, unlike 16S (bacteria), which (currently) can rarely identify ASVs past the genus level. Classifiers, including the UNITE classifier used in this study, are rather conservative, and will usually not give a species-level identification unless confident in that assignment. Due to the methodology used in this study (paired-end sequencing for longer read lengths, ITS-specific denoising that does not trim all sequences to the same length, and the use of a dynamic classifier that can improve the accuracy of taxonomic assignment), many of the ASVs in this study were identified to the species-level with >97% confidence. We note that there were other Hanseniaspora spp. also identified to the species-level that were present in lower relative abundance in this study, including H. uvarum, H. guilliermondii, and H. valbyensis (see ‘fungal community’ tab in the ‘manuscript-data’ Excel file on OSF - https://osf.io/j7rx8/). Based on the reviewer’s inquiry, we also confirmed the species identity of the ASVs identified as H. osmophila via NCBI BLAST. Furthermore, we note that there were a number of ASVs that could not be confidently identified to the species level using our classifier - those ASVs (such as Penicillium sp., Candida sp., and others) were identified to the most accurate taxonomic level available. Based on this question, we have added a taxonomy.qzv file to our supplemental information (https://osf.io/j7rx8/), which contains more information regarding the ASV sequences in this study, their taxonomic classification, and the confidence of the taxonomic assignments. To view this document, simply visit view.qiime2.org and drag and drop the taxonomy.qzv file into the box.

---

## [Decision Letter · Decision Letter 2]

29 Dec 2020

An indigenous Saccharomyces uvarum population with high genetic diversity dominates uninoculated Chardonnay fermentations at a Canadian winery

PONE-D-19-30860R2

Dear Dr. Morgan,

We’re pleased to inform you that your manuscript has been judged scientifically suitable for publication and will be formally accepted for publication once it meets all outstanding technical requirements.

Kind regards,

Luca Cocolin

Academic Editor

PLOS ONE

Additional Editor Comments (optional):

Reviewers' comments:

Reviewer's Responses to Questions

**Comments to the Author**

1. If the authors have adequately addressed your comments raised in a previous round of review and you feel that this manuscript is now acceptable for publication, you may indicate that here to bypass the “Comments to the Author” section, enter your conflict of interest statement in the “Confidential to Editor” section, and submit your "Accept" recommendation.

Reviewer #1: All comments have been addressed

2. Is the manuscript technically sound, and do the data support the conclusions?

Reviewer #1: Yes

3. Has the statistical analysis been performed appropriately and rigorously? 

Reviewer #1: No

4. Have the authors made all data underlying the findings in their manuscript fully available?

Reviewer #1: Yes

5. Is the manuscript presented in an intelligible fashion and written in standard English?

Reviewer #1: Yes

6. Review Comments to the Author

Reviewer #1: This new version answers to my concerns in a satisfactory manner. I think that this is a nice study.

7. PLOS authors have the option to publish the peer review history of their article (what does this mean?). If published, this will include your full peer review and any attached files.

Reviewer #1: **Yes: **Jean-Luc Legras

---

## [Editor Report · Acceptance letter]

25 Jan 2021

PONE-D-19-30860R2 

An indigenous *Saccharomyces uvarum* population with high genetic diversity dominates uninoculated Chardonnay fermentations at a Canadian winery 

Dear Dr. Morgan:

I'm pleased to inform you that your manuscript has been deemed suitable for publication in PLOS ONE. Congratulations! Your manuscript is now with our production department. 

Kind regards, 

on behalf of

Dr. Luca Cocolin 

Academic Editor

PLOS ONE